# Stepping on the Edge:
# Curvature Aware Learning Rate Tuners

**Vincent Roulet**[*]
Google DeepMind
vroulet@google.com

**Atish Agarwala**[*]
Google DeepMind
thetish@google.com

**Jean-Bastien Grill**
Google DeepMind
jbgrill@google.com

**Grzegorz Swirszcz**
Google DeepMind
swirszcz@google.com

**Mathieu Blondel**
Google DeepMind
mblondel@google.com

**Fabian Pedregosa**
Google DeepMind
pedregosa@google.com

## Abstract

Curvature information – particularly, the largest eigenvalue of the loss Hessian, known as the sharpness – often forms the basis for learning rate tuners. However, recent work has shown that the curvature information undergoes complex dynamics during training, going from a phase of increasing sharpness to eventual stabilization. We analyze the closed-loop feedback effect between learning rate tuning and curvature. We find that classical learning rate tuners may yield greater one-step loss reduction, yet they ultimately underperform in the long term when compared to constant learning rates in the full batch regime. These models break the stabilization of the sharpness, which we explain using a simplified model of the joint dynamics of the learning rate and the curvature. To further investigate these effects, we introduce a new learning rate tuning method, Curvature Dynamics Aware Tuning (CDAT), which prioritizes long term curvature stabilization over instantaneous progress on the objective. In the full batch regime, CDAT shows behavior akin to prefixed warm-up schedules on deep learning objectives, outperforming tuned constant learning rates. In the mini batch regime, we observe that stochasticity introduces confounding effects that explain the previous success of some learning rate tuners at appropriate batch sizes. Our findings highlight the critical role of understanding the joint dynamics of the learning rate and curvature, beyond greedy minimization, to diagnose failures and design effective adaptive learning rate tuners.

## 1 Introduction

The learning rate, a.k.a. stepsize, is the main hyperparameter controlling the efficiency and stability of gradient-based training of deep neural networks. The learning rate is typically adjusted through a predetermined schedule – often consisting of a warm-up phase, where the learning rate is gradually increased to a peak, followed by an annealing phase, where it is decreased to zero (Goyal et al., 2017; Loshchilov and Hutter, 2016). Tuning the shape of the schedule (warm-up time, peak learning rate, decay scale and shape) is essential for good performance. Despite recent efforts to understand their effectiveness, the optimal shape of these schedules remains an area of active research (Liu et al., 2020; Shi et al., 2023). The cost of tuning these schedules has led to interest in automatic selection of these hyperparameters with *learning rate tuners* - methods which aim to automatically adjust the learning rate through training.

---

[*]Equal contribution.

38th Conference on Neural Information Processing Systems (NeurIPS 2024).

These methods have roots in traditional optimization theory, including inexact linesearch with Armijo-Goldstein criterion (Armijo, 1966; Nocedal and Wright, 1999) and Polyak stepsizes (Polyak, 1964), which select the learning rate via estimates of the gap to optimality of the objective. The Armijo-Goldstein criterion is a crucial component of popular full-batch convex optimizers, such as L-BFGS (Liu and Nocedal, 1989). Recent efforts have adapted linesearches to stochastic optimization, with some partial empirical successes and with some approaches offering convergence guarantees (Galli et al., 2023; Mutschler and Zell, 2020; Vaswani et al., 2019). Similar efforts have been made for Polyak stepsizes (Berrada et al., 2020; Loizou et al., 2021), in addition to new methods which combine distance to optimality with online learning convergence bounds (Cutkosky et al., 2023; Defazio and Mishchenko, 2023; Ivgi et al., 2023; Mishchenko and Defazio, 2023).

Classically-inspired methods, however, have generally struggled to gain traction in deep learning. This is partly due to their design, which prioritizes convex, Lipschitz-continuous, and/or smooth (Lipschitz-continuous gradients) objectives. In contrast, the loss landscape of deep networks is known to be non-convex (Li et al., 2018), and non-Lipschitz continuous (Hochreiter et al., 2001). Moreover, non-linear models, especially neural networks, will commonly undergo dramatic changes in geometry during training (Arora et al., 2022; Jastrzębski et al., 2019; Jastrzebski et al., 2020; Kalra et al., 2023; Kopitkov and Indelman, 2020; Lewkowycz et al., 2020; Wu et al., 2018). In particular, most models undergo a phase of *progressive sharpening* - where the sharpness, the largest eigenvalue of the Hessian, increases during training (Cohen et al., 2021). These potentially detrimental effects are mitigated by non-linear stabilization arising from the discreteness of the dynamics – namely, the *edge of stability* (EOS) phenomenon (Cohen et al., 2021). This causes large Hessian eigenvalues to stabilize at the critical value for a given learning rate in an equivalent smooth setting (for example, max Hessian eigenvalue stabilizes at $\lambda_{\max} = 2/\eta$ for learning rate $\eta$) (Cohen et al., 2023, 2021). The early training time behavior corresponds to the regime where there is the most feature learning (Cohen et al., 2023, 2021), and is the main focus of this work; at late times, the large eigenvalues of the Hessian usually drop below the edge of stability. Gilmer et al. (2022) considered EOS stabilization as a leading candidate for the necessity of the warm-up procedure; as the learning rate $\eta$ increases, $\lambda_{\max}$ is effectively annealed.

This raises some natural questions. *How do these sharpness dynamics affect the performance of learning rate tuners? What insights can we gain to design better tuners for deep learning?* Our work takes a first step at answering these questions, starting with a study of some classical learning rate tuners: a linesearch ensuring sufficient decrease and an approximately greedy method that minimizes a quadratic approximation of the objective. Specifically, we find the following.

- We empirically observe that classical learning rate tuners qualitatively underperform their constant learning rate counterparts across several deep learning benchmarks, in the full batch regime, for which these methods were originally designed.
- Our empirical analysis of curvature dynamics reveals that classical learning rate tuners generally undershoot the edge of stability. This undershooting creates a snowball effect of ever-increasing sharpness and ever-decreasing learning rates.
- We propose a theoretical model that effectively captures these empirically observed failures.

Our analysis suggests that stabilizing the sharpness may be a more important goal for the long-term success of training, compared to greedily optimizing the objective. To explore this idea, we propose the Curvature Dynamics Aware Tuning (CDAT) method, which dynamically drives the learning rate to the EOS. In our exploration, we find the following.

- We observe empirically that the proposed learning rate tuner can outperform fine-tuned constant learning rate counterparts in a full batch regime.
- We analyze the sharpness dynamics induced by CDAT in these examples and observe that the progressive sharpening is mitigated by the tuner, increasing learning rates at early times before stabilizing, akin to an automatic warm-up schedule.
- We propose a theoretical model that clarifies the dynamical mechanisms by which CDAT maintains proximity to the EOS, while highlighting the limitations of existing models of curvature dynamics.

Our work suggests that the design of learning rate tuners benefits from exploiting curvature stabilization rather than focusing on loss decrease. The introduction of simple learning rate tuners can also refine our understanding of sharpness dynamics through feedback loop effects. Additional experiments and experimental details are presented in Appendix B and Appendix C respectively.

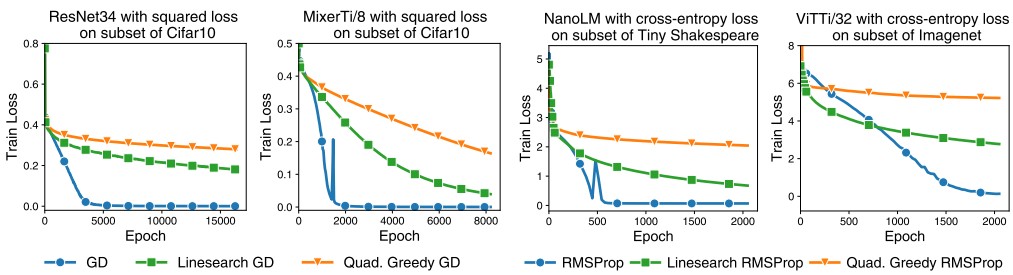

Figure 1: **Simple learning rates tuners qualitatively underperform their constant learning rate counterparts.** Gradient descent or RMSProp with a tuned constant learning rate versus self-tuned gradient descent by a linesearch method (1), or a quadratically greedy rule (3) on various datasets, architectures and losses in a full batch regime. The linesearch may perform better at early times but stalls in the long term.

## 2 The Interplay Between Learning Rate Tuners and Curvature Dynamics

A leitmotif in the design of learning rate tuners has been to select the learning rate to ensure a maximal or sufficient decrease of the objective at each iteration. We focus here on two canonical examples. Polyak stepsizes and hyper-gradient descent are also briefly examined in Appendix B, Fig. 13.

### 2.1 Canonical learning rate tuners failures in deep learning

The first classical approach we consider is a **linesearch** (ls) method that selects the learning rate $\eta$ such that the objective $f$ satisfies a certain decrease criterion (Armijo, 1966; Nocedal and Wright, 1999). Formally, given current parameters $w_t$ and an update direction $u_t$, the learning rate $\eta_t^{\text{ls}}$ is chosen such that

$$f(w_t + \eta_t^{\text{ls}} u_t) \leq f(w_t) + c\, \eta_t^{\text{ls}} u_t^\top \nabla f(w_t)\,. \tag{1}$$

This rule assumes that $u_t$ is a descent direction ($\nabla f(w_t)^\top u_t < 0$), which ensures the existence of a learning rate satisfying (1). This holds true for simple Gradient Descent (GD) or preconditioned variants like RMSProp (Hinton et al., 2012). In the criterion (1), $c$ is usually a small constant set to $10^{-4}$ or 0. A valid learning rate is searched with a usual backtracking linesearch (Appendix C).

The second method we consider involves selecting the learning rate at each iteration to minimize a quadratic approximation of the objective. Formally, the objective $f$ at parameters $w_t$ can be approximated along an update direction $u_t$ by a quadratic approximation $q_f$ as

$$f(w_t + \eta u_t) \approx q_f(\eta; w_t, u_t) := f(w_t) + \eta \nabla f(w_t)^\top u_t + \frac{1}{2}\eta^2 u_t^\top \nabla^2 f(w_t) u_t. \tag{2}$$

Provided that this quadratic approximation is strongly convex in $\eta$ ($u_t^\top \nabla^2 f(w_t) u_t > 0$), the minimum of the quadratic approximation $q_f(\eta; w_t, u_t)$ is reached for the **quadratically greedy** (qg) learning rate $\eta^{\text{qg}}$ given by

$$\eta_t^{\text{qg}} = \frac{-\nabla f(w_t)^\top u_t}{u_t^\top \nabla^2 f(w_t) u_t}\,. \tag{3}$$

Setting the learning rate by minimizing the quadratic approximations (3) is a simple intuitive idea studied for example by Schaul et al. (2013), Martens and Grosse (2015, Section 6.4). This approach as well as linesearches are effective on simple linear problems (Fig. 2). While their rationale originates in non-stochastic optimization, they have been analyzed in the context of stochastic optimization for deep learning (Schaul et al., 2013; Vaswani et al., 2019).

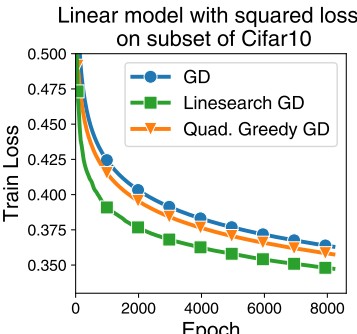

Figure 2: Classical learning rate tuners can be effective on linear models.

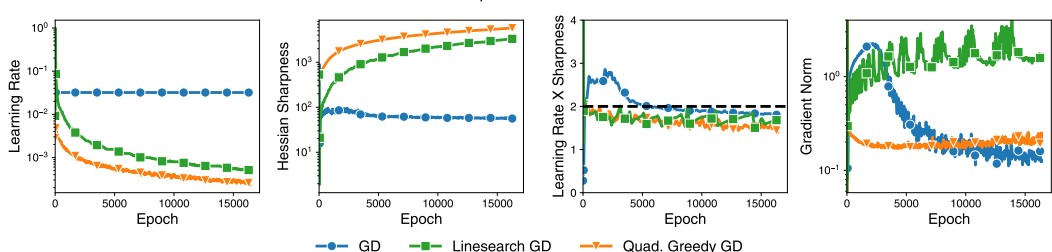

Figure 3: **Classical learning rate tuners can undershoot the edge of stability.** Learning rate, sharpness, their product, and the gradient norm evolution of a constant learning rate and learning rate tuners, full batch gradient descent. Learning rate decreases by 3 orders of magnitude for tuners (1st panel) while sharpness increases (2nd panel). Their product remains relatively steady, just below the edge of stability (3rd panel). The gradient norm increases by less than a factor of 10, consistent with slow training at late times (4th panel).

## 2.2 Analyzing learning rate tuners through curvature dynamics

**Full batch regime.** We revisit the performance of the learning tuners presented in Section 2.1 in the full batch regime on deep learning problems in Fig. 1. As demonstrated in Fig. 1, a linesearch (1) or the quadratically greedy rule (3) qualitatively underperform their constant learning rate counterpart in the deep learning benchmarks considered. Notably, all these results are obtained despite being in a full batch regime, for which these methods are originally designed. To understand the failures of these approaches, we consider several measures presented in Fig. 3 (see also Fig. 12).

First, we observe a consistent decrease in the chosen learning rate over time, spanning several orders of magnitude (1st panel of Fig. 3). This is surprising, as none of these approaches explicitly encode a decreasing learning rate mechanism. Specifically, the linesearch always initiates its search with a guess larger than the previously selected learning rate (see Appendix C for implementation details). Decreasing learning rates are theoretically optimal for non-smooth objectives (Nesterov et al., 2018), such as the ones induced by using the ReLU activation; however in our example, the gradient norm does not increase beyond one order of magnitude (4th panel of Fig. 3). This suggests both that an increase in gradient norm is not the primary cause of learning rate decrease, and also explains why the learning rate decrease is correlated with slower progress on the training loss.

Following the work of Cohen et al. (2021), we analyze the dynamics of the sharpness, that is the largest eigenvalue of the Hessian, $\lambda_{\max}(\nabla^2 f(w_t))$. In the 2nd panel of Fig. 3, we observe that while sharpness stabilizes for gradient descent, it does not exhibit the same behavior for the considered learning rate tuners. By plotting the product of the learning rate $\eta_t$ and the sharpness (3rd panel of Fig. 3), we find that this product can exceed the stability threshold of 2, eventually stabilizing below this threshold for constant learning rate gradient descent. In contrast, for the learning rate tuners, this product neither surpasses the stability threshold nor stabilizes around 2 in the long run. Therefore, these classical learning rate tuners do not operate at the edge of stability.

From a theoretical perspective, objectives are typically classified as either smooth or non-smooth. Smooth objectives have gradients that are Lipschitz-continuous, at least locally around any point. Non-smooth objectives, on the other hand, may contain points with kinks (non-differentiable points). However, this taxonomy might not fully capture the curvature dynamics observed by Cohen et al. (2023, 2021) for constant learning rates, and in Fig. 1 for the classical learning rate tuners. In particular, the concept of smoothness might not be entirely relevant in the context of deep learning, where its local estimate (the spectral norm of the Hessian, also known as sharpness) can continue to increase throughout training. To push the limits of classical smoothness assumptions, we consider in Section 3 a learning rate tuner that propels the optimizer at the edge of stability or above, a regime that usual smoothness assumptions would theoretically prohibit.

**Mini-batch regime.** The results presented in Fig. 1 in the full batch regime *do not contradict* the success of linesearches at medium batch size observed by Vaswani et al. (2019) in the stochastic regime. This observation is illustrated in Fig. 14, and was previously reported by Roulet et al. (2023). We simply point out that the success of linesearches observed by Vaswani et al. (2019) may not be entirely attributable to the method's original rationale.

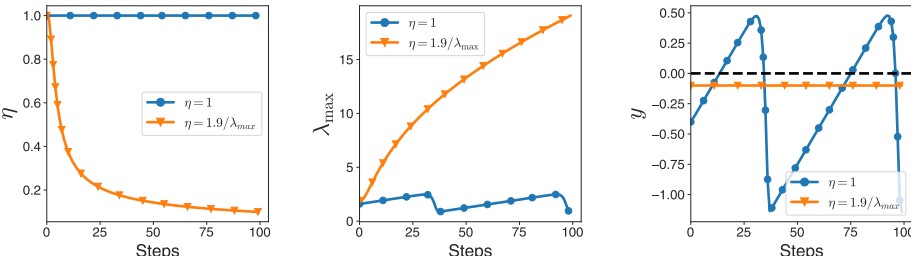

Figure 4: **The poor performance of classical learning rate tuners, understood in a simplified model.** The dynamics of learning rate $\eta$, sharpness $\lambda_{\max}$, and normalized centered sharpness $y = \eta\lambda_{\max} - 2$ are examined in the simplified model (5). With a constant $\eta$, $\lambda_{\max}$ stabilizes and $y$ oscillates around 0 (blue). Classical learning rate tuners often quickly equilibrate around $y_t = -\epsilon$, which we model using $\eta = 1.9\lambda_{\max}$ (orange). This equilibration of $y$ away from zero prevents stabilization in $\lambda_{\max}$, leading to an increase in $\lambda_{\max}$, and a corresponding decrease in $\eta$.

The actual success of linesearches in a stochastic regime may instead be explained by the attenuated progressive sharpening observed in such a regime (Agarwala and Pennington, 2024; Cohen et al., 2021; Jastrzębski et al., 2017). Moreover, linesearches applied to mini-batches tend to select larger learning rates than they would in a full-batch regime (Mutschler and Zell, 2020) potentially allowing them to avoid undershooting the full objective's edge of stability.

## 2.3 Theoretical analysis

The sharpening effects can be understood theoretically. Previous work has shown that the stabilization provided by EOS is due to non-linear interaction between the component of the gradient in the largest eigendirection, and the dynamics of the largest eigenvalues themselves (Agarwala et al., 2023; Damian et al., 2023). We can use these analyses to understand why there is no stabilization for some classical learning rate tuners.

We start with the model from Damian et al. (2023), which focuses on the dynamics in the largest eigendirection of the Hessian. We consider a unique eigenvector for simplicity; we don't observe degeneracy in the eigenspace of the largest eigenvalue in any practical models. Given an objective $f$ parameterized by parameters $w_t$, let $\lambda_t$ be the largest eigenvalue of the Hessian $\nabla^2 f(w_t)$, i.e., $\lambda_t := \lambda(w_t) := \lambda_{\max}(\nabla^2 f(w_t))$. Let $v$ be its normalized eigenvector; the model assumes slow eigenvector change, so it is treated as a fixed direction. The joint dynamics of $\lambda_t$ and the projection $x_t := v^\top w_t$ can then be written as

$$x_{t+1} = (1 - \eta_t\lambda_t)x_t, \quad \lambda_{t+1} = \eta_t(a - bx_t^2) + \lambda_t. \qquad (4)$$

Here, $a := -\nabla\lambda(w)^\top\nabla f(w)$ corresponds to the instantaneous change of $\lambda$ along the negative gradient (the update direction), and $b := \|\nabla\lambda(w)\|^2$ encodes the non-linear negative feedback between $x_t$ and $\lambda_t$. Both $a$ and $b$ are considered constant along iterations. These equations are derived by Damian et al. (2023) using a Taylor expansion of the iterates combined with a coupling argument. We provide intuition for the model in Appendix A.1.

In the original model, the learning rate $\eta_t$ is also fixed to $\eta$. This leads to the following dynamics: while $\eta\lambda_t < 2$, the magnitude of $x_t$ decreases. This, in turn, leads to an increase in $\lambda_t$. Eventually, $\eta\lambda_t > 2$ and $|x_t|$ increases. This eventually leads to the $bx_t^2$ term becoming large, which decreases $\lambda_t$. There is a range of learning rates over which this dynamic leads to quasi-stable oscillations of $\lambda_t$ around the edge of stability value $2/\eta$ (Fig. 4, blue curves).

When using a learning rate tuner, $\eta_t$ is also a dynamical variable. This introduces the additional complication of a shifting edge of stability. Therefore, it is advantageous to analyze the dynamical system using normalized variables (Agarwala et al., 2023). We define $y_t := \eta_t\lambda_t - 2$, where $y = 0$ corresponds to the EOS, and $p_t := x_t^2$. This gives us the dynamical equations (Appendix A.2)

$$p_{t+1} = (1 + y_t)^2 p_t, \quad y_{t+1} = \eta_{t+1}\left[\eta_t\left(a - bp_t\right)\right] + \left(\frac{\eta_{t+1}}{\eta_t}\right)y_t + 2\left[\frac{\eta_{t+1}}{\eta_t} - 1\right]. \qquad (5)$$

We must then supply a rule for $\eta_{t+1}$. In Fig. 3, we observed that in the full batch setting, the learning rate multiplied by the sharpness appears to quickly approach a threshold of $2 - \epsilon$ (corresponding to $y = -\epsilon$), and then varies slowly below the EOS threshold.

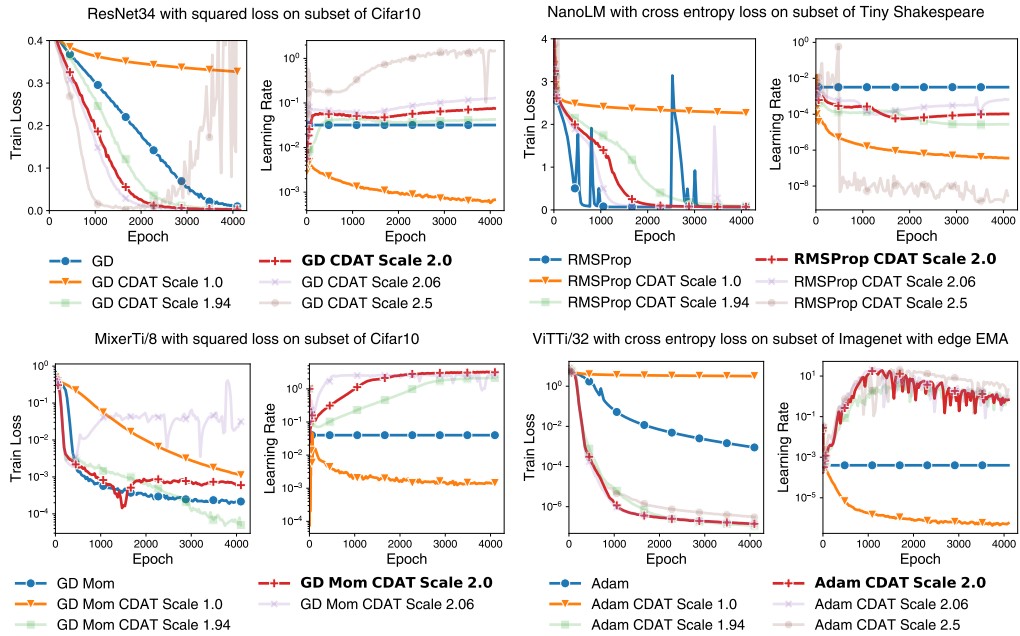

Figure 5: **Enforcing optimizers to stay on edge** ($\sigma = 2.0$) **improves performance over greedy approximation** ($\sigma = 1.0$). Train loss and learning rate behaviors for fine-tuned optimizers vs self-tuned counterparts with CDAT on various datasets, architectures, losses in a full batch regime. Tuning the learning rate "on edge" ($\sigma \approx 2$) improves performance over greedy tuning ($\sigma = 1$) as well as constant learning rate.

We model the varying learning rate as

$$\eta_t := 2(1 - \epsilon)/\lambda_t \,. \tag{6}$$

This maintains $y_t = -\epsilon$. Notably, this schedule was explicitly proposed by Cohen et al. (2021) (see also Fig. 15). In this regime, $p_t$ decreases monotonically, aligning with the original goal of these methods to decrease the loss (Fig. 10). However, this eliminates feedback for controlling the increase in $\lambda_t$, resulting in significant progressive sharpening (Fig. 4, orange curve).

Consequently, when attempting to enforce monotonicity, learning rate tuners may inadvertently disrupt the non-linear stabilization that makes gradient descent robust and effective for training deep neural networks. Continually undershooting the EOS triggers a snowball effect of decreasing learning rate and increasing sharpness. If there is no corresponding increase in gradient norms, this causes optimization to slow down.

The poor performance of the classical learning rate tuners in Fig. 1 therefore appear strongly correlated with their tendency to *undershoot* the edge of stability in the normalized sharpness coordinate $y$. In the following, we focus on understanding tuners that prioritize training at or near the edge of stability.

## 3 Optimizing on the Edge of Stability

Based on our observations in Section 2, we design learning rate tuners that position the underlying optimizer **on the edge** of stability ($y = 0$). We analyze a tuner capable of operating both slightly below and slightly above the EOS in order to exploit nonlinear stabilization.

Formally, we investigate a generalization of the quadratically greedy rule from Section 2, which sought $\eta_t$ to minimize the quadratic approximation $q_f$ in (2). We instead choose the learning rate to be *on edge* by seeking the largest value of $\eta$ such that $q_f$ is smaller or equal to the original value of $f$,

$$\eta_t^{\text{oe}} := \max\{\eta \geq 0 : q_f(\eta; w_t, u_t) \leq f(w_t)\} = -2\frac{\nabla f(w_t)^\top u_t}{u_t^\top \nabla^2 f(w_t) u_t} \,, \tag{7}$$

where the last formula holds provided that $u_t^\top \nabla^2 f(w_t) u_t > 0$ (convex quadratic) and $\nabla f(w_t)^\top u_t < 0$ ($u_t$ is a descent direction).

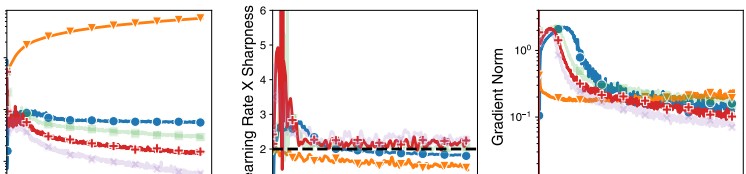

Figure 6: **Optimizing on edge induces different curvature dynamics.** Sharpness, product between learning rate and sharpness, and gradient norm evolutions for gradient descent with CDAT. By putting the learning rate on edge ($\sigma \approx 2$), the sharpness does not ever increase and actually decreases slightly over time. GD with CDAT operates slightly above the edge constantly during training. Its gradient norm evolution is akin to a fine-tuned constant learning rate baseline.

For $u_t = -\nabla f(w_t)$, and if $-\nabla f(w_t)$ is aligned with the eigendirection $v_{\max}$ associated with the largest eigenvalue $\lambda_{\max}$ of $H$, we recover the familiar $\eta_t^{\text{oe}} = 2/\lambda_{\max}$. Note however, that contrarily to using directly $\eta_t = 2/\lambda_{\max}$, the on-edge rule can naturally take into account the alignment with $v_{\max}$ (see Fig. 15). We note that we recover the edge of stability even when the updates are given by the gradient multiplied by a preconditioner, e.g. $u_t = -P^{-1}\nabla f(w_t)$ for a matrix $P$. In this case, we have $-u^\top H u/u^\top g = g^\top P^{-1} H P^{-1} g/g^\top P^{-1} g$ for $H = \nabla^2 f(w_t)$, $g = \nabla f(w_t)$. This is maximized when $P^{-1/2}g$ lies in the largest eigendirection of the PSD matrix $\tilde{H} \equiv P^{-1/2} H P^{-1/2}$, which for $\sigma = 2$ gives us the learning rate $\eta_t = 2/\tilde{\lambda}_{\max}$, where $\tilde{\lambda}_{\max}$ is the maximum eigenvalue of $\tilde{H}$. This is exactly the edge of stability for adaptive methods (Cohen et al., 2023).

We note that the only difference between this and the quadratically greedy rule is a factor of 2 in the numerator. Inspired by this observation, and with an eye towards robustness, we define our *Curvature Dynamics Aware Tuning* (CDAT) rule by:

$$\eta_t^{\text{cdat}} = \sigma \frac{n_t}{d_t}, \quad \text{for } n_t = \max\{-\nabla f(w_t)^\top u_t, 0\}, \; d_t = |u_t^\top \nabla^2 f(w_t) u_t| + \varepsilon. \tag{8}$$

The scaling factor $\sigma$ lets us interpolate between greedy ($\sigma = 1$) and on-edge ($\sigma = 2$). We are most interested in the behavior near $\sigma = 2$, (also studied in Rosca et al. (2023)). In (8), the max function takes care of the case where $u_t$ is an ascent direction ($\nabla f(w_t)^\top u_t > 0$), the absolute value takes care of cases where the objective has negative curvature in the update directions (see Appendix C for additional justification), and we simply set $\varepsilon = 0$ as we always observed non-negligible positive curvature. The definitions of the numerator $n_t$ and the denominator $d_t$ allow for the possibility of exponential moving averages (EMA) of each quantity such as $\tilde{n}_{t+1} = (1 - \beta_{\text{cdat}})n_t + \beta_{\text{cdat}}\tilde{n}_t$ for $\beta_{\text{cdat}}$ referred to as the CDAT EMA parameter thereafter. We observed that smoothing the estimates of $n_t$ and $d_t$ by an EMA is particularly relevant when the updates are themselves defined through an exponential moving average as in Adam, or when using the proposed rule in a stochastic setting.

CDAT has two major advantages: it is sensitive to information from all eigenvalues of $\nabla^2 f(w_t)$, and it depends on updates $u_t$ coming from any base optimizer. We will take advantage of these properties to explore the behavior of "on edge" optimization in a variety of settings.

### 3.1 On edge optimizers in practice

**Full batch regime.** Fig. 5 presents results for training with CDAT across various optimizers, architectures, datasets, and losses. Overall, selecting the learning rate to be on edge ($\sigma = 2$) is on par with or better than a fine-tuned constant learning rate and is always better than a quadratically greedy approach ($\sigma = 1$). This observation holds even though the quadratically greedy rule ensures larger instantaneous decrease (Fig. 16). One notes that targeting slightly above the edge ($\sigma = 2.0625$) provides even better performance than the on edge rule ($\sigma = 2$) on all examples except the MLP Mixer on CIFAR10. However, targeting higher above the edge ($\sigma = 2.5$) generally gives diverging results in the short or long terms. To integrate the proposed rule with the Adam optimizer, we also observed that the estimation of the curvatures through $n_t$, $d_t$ in (8) was necessary.

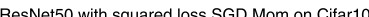

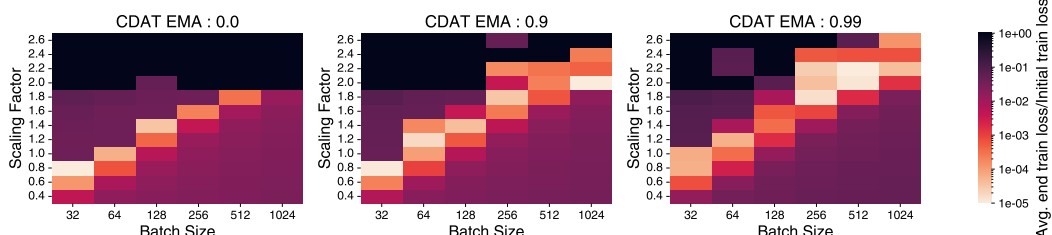

Figure 7: **Stochasticity shifts the optimal scaling.** Normalized performance of gradient descent with momentum equipped with CDAT in a stochastic regime with varying batch sizes. In a mini-batch regime, the optimal scale decreases as the batch size decreases. Using an exponential moving average smooths out the performance of the CDAT rule over batch sizes.

Remarkably, all choices around the edge $(1.9375, 2.0, 20625)$ show a progressive increase of the learning rate that results generally in a better performance than the constant learning rate counterparts, except for RMSProp on the NanoLM experiment. The increasing learning rate behavior is akin to the warm-up phase generally hard-coded by a scheduler. In Fig. 19, we observe that the CDAT rule displays similar behavior as warm-up schedules, yet it may not fully capture the benefits of prefixed schedules.

In Fig. 6, we analyze the dynamics of the curvature when optimizing on edge. We observe that the sharpness can be pushed to reduce over the iterations (1st panel of Fig. 6). The CDAT rule may operate constantly slightly above the edge (2nd panel of Fig. 6). By reducing the sharpness, the algorithm may be able to take larger stepsizes and converge faster. Sensitivity to architecure's width and depth, as well as weight decay, are also analyzed in Fig. 18.

**Mini batch regime.**  The CDAT rule can be used in a stochastic regime by replacing $f$ in (8) by its stochastic counterpart $f^{(m_t)}$ on a mini-batch $m_t$. However, two difficulties may arise.

First, the on edge rule is motivated by the sharpening effects of the overall objective, which can be overestimated or underestimated by a single mini-batch. Previous work shows that the trace of the Hessian may best capture the sharpening and stabilization effects in a stochastic regime (Agarwala and Pennington, 2024; Wu and Su, 2023); it is unclear what function of the Hessian spectrum, the CDAT rule captures in the stochastic regime. As a result the optimal scaling factor may vary with the mini-batch. In Fig. 7, we observe that the optimal scaling of the on-edge rule is proportional to the batch size up to some size. In particular, at specific batch sizes, we observe that the greedy rule ($\sigma = 1$) outperforms the on-edge rule. This result is consistent with the good performance of linesearches or greedy rules in a mini-batch regime previously mentioned and observed in Fig. 14. We also observe in Fig. 7, that integrating an EMA into the estimation of the edge in (8) smooth out the selection of the optimal scaling factor.

Finally, the sharpening effects are known to be generally mitigated in the stochastic regime (Agarwala and Pennington, 2024; Cohen et al., 2021; Jastrzębski et al., 2017). The benefits of the on edge rule appear also subdued in this regime (Fig. 8, Fig. 20, Fig. 21).

## 3.2  Modeling CDAT dynamics

The classical optimization framework is insufficient to fully explain the benefits of CDAT. For example, on a convex quadratic objective, $\sigma = 1$ is the optimal choice, and $\sigma > 2$ results (in the worst case) in a divergent algorithm. However, we can use a simplified model to begin understanding the joint dynamics of the learning rate and sharpness under CDAT.

We approximate the gradients around a stationary point $w_\star$, where $\nabla f(w_\star) = 0$, as $\nabla f(w_t) \approx H\bar{w}_t$ for $\bar{w}_t := w_t - w_\star$, and $H$ being a symmetric matrix. In this scenario, the learning rate given by CDAT is $\eta_t^{\text{cdat}} = \sigma(\bar{w}_t^\top H^2 \bar{w}_t)/(\bar{w}_t^\top H^3 \bar{w}_t)$. Consider the case where $H$ has two eigenvalues $\lambda$ and $\nu$, with $\lambda > \nu \geq 0$. In this case the CDAT learning rate can be written as

$$\eta_t^{\text{cdat}} = \sigma \frac{\lambda^2 p_t + \nu^2 g_t}{\lambda^3 p_t + \nu^3 g_t} = \sigma \frac{\lambda^2 p_t/g_t + \nu^2}{\lambda^3 p_t/g_t + \nu^3} . \tag{9}$$

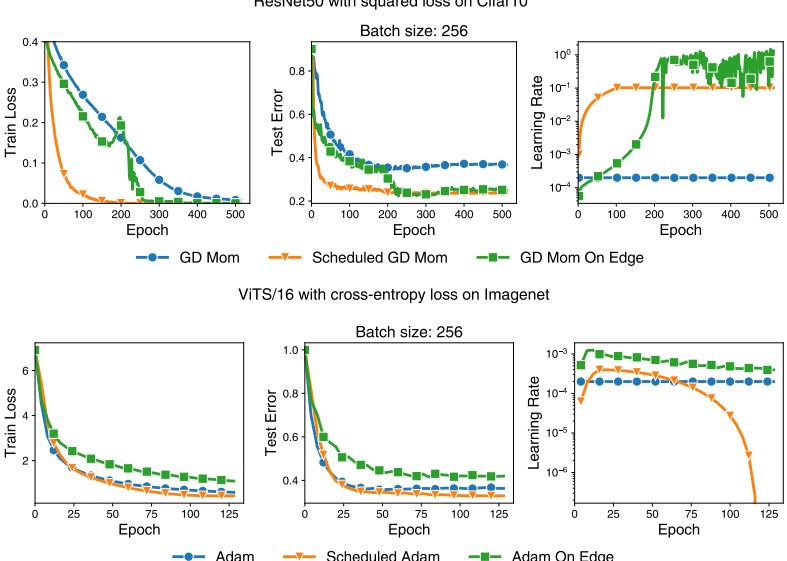

Figure 8: **The performance of CDAT is subdued in the stochastic regime.** Fine-tuned constant, scheduled, and self-tuned with CDAT learning rates in a stochastic regime. In a stochastic regime, CDAT can also exhibit a form of learning rate warm-up (top figure). However, the interplay between sharpening and learning rate are known to be mitigated in a stochastic regime which may explain the underperformance of CDAT in this regime (bottom figure).

Here $p_t, g_t$ are the projections $p_t := (\bar{w}_t^\top v)^2, g_t := (\bar{w}_t^\top v_\perp)^2$, respectively onto the eigendirections $v$ and $v_\perp$ associated with $\lambda, \nu$. Therefore, $\eta_t^{\text{cdat}}$ interpolates between its minimum value $\sigma/\lambda$ to the larger value $\sigma/\nu$, depending on the alignment ratio $p_t/g_t$. For $2\nu/\lambda < \sigma < 2$, this rule can achieve learning rates both above and below the EOS.

We can gain additional insight by modeling a dynamical $\lambda_t$, extending the model of Section 2.3. While model (5) captures the dynamics in the largest eigendirection $v$, here we aim to model the dynamics in the orthogonal subspace. To simplify, we consider the eigendirections $v, v_\perp$, and small eigenvalue $\nu$ fixed. We then model the gradients as $\nabla f(w_t) \approx H_t \bar{w}_t$ with $H_t = \lambda_t v v^\top + \nu v_\perp v_\perp^\top$. If we update $w$ in the direction $v_\perp$ using gradient descent on $\nu g_t$, we obtain the following dynamical system describing the CDAT learning rate tuner:

$$\eta_{t+1} = \sigma \frac{\lambda_t^2 p_t + \nu^2 g_t}{\lambda_t^3 p_t + \nu^3 g_t}, \quad g_{t+1} = (1 - \eta_t \nu_t)^2 g_t, \quad p_{t+1} = (1 + y_t)^2 p_t. \tag{10}$$

Combining this with the update rule for $y_t$ given in (5) completes the model.

There are two important regimes of behavior in this model. First, if $y_t > 0$, $p_t$ will increase and eventually $y_t$ will decrease as in the normal EOS case. If $y_t < 0$, the key threshold is $y_t < -\eta_t \nu_t$. In this case, the ratio $p_t/g_t$ *decreases* - leading to an increase in $\eta_t$ according to the on edge rule. If $a - b p_t > 0$ (as it is if $p_t$ has become small due to $y_t < 0$), then we see from (5) that this leads to an *increase* in $y_t$. This suggests that CDAT has a tendency to push $y_t$ closer to the EOS – sending $y$ towards 0 if the learning rate is driven by the eigendirections corresponding to smaller eigenvalues.

Numerical simulations on this model (Fig. 9) suggest that this effect can indeed cause remarkably small values of $y$ (3rd panel of Fig. 9). We emphasize that this is due to the *joint dynamics* of $\eta_t$ (induced by the learning rate tuner), and $\lambda_t, p_t$, and $g_t$ (induced by GD). There are also important limitations in this model's ability to fully explain CDAT's behavior. For example, the model predicts runaway sharpening for $\sigma < 2$ (2nd panel of Fig. 9), and divergence for $\sigma > 2$. In practice, we saw a range of stable and useful settings for scale centered around 2. This modeling limitation likely stems from neglecting the dynamics orthogonal to $v$ as well as higher-order terms, which empirically tend to stabilize EOS dynamics (Agarwala et al., 2023).

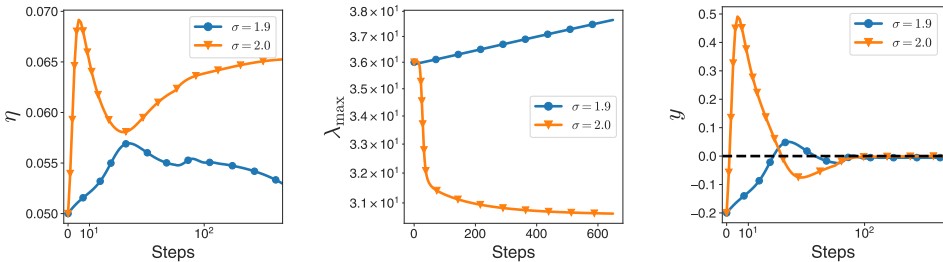

Figure 9: **A simple model partially captures the benefits induced by the proposed CDAT rule.** Dynamics of theoretical model of CDAT (10). For $\sigma = 2$, feedback stabilizes $y$ close to the EOS ($y = 0$), which stabilizes $\lambda_{\max}$ (orange). For $\sigma = 2 - \epsilon$ and small $\epsilon$ (blue, $\epsilon = 0.1$), model predicts that $\lambda_{\max}$ slowly grows (middle), but predicts that $y$ stabilizes to a value $-\epsilon \ll y_t < 0$ (right).

## 4 Conclusion and Future Directions

**Summary.** Our empirical results showed that simple linesearches and approximate greedy learning rate tuners underperform constant learning rate approaches in the full batch regime – despite being better on individual steps. The idea that "locally greedy" methods perform poorly on long time scales has been shown in other settings as well, including evolutionary dynamics Agarwala and Fisher (2019). Our experiments and theoretical work suggest the failure of these classical tuners is due to the fact that they suppress the feedback which stabilizes sharpness in the fixed learning rate setting. As the sharpness increases, tuners are forced to take smaller steps, which ends up leading to slower learning.

We find, in contrast, that prioritizing stability of the sharpness yields tangible benefits. Our CDAT method pushes the network towards the edge of stability via a dynamically driven process. It also naturally displays some form of progressive increase of the learning rate akin to prefixed warm-up schedules. CDAT also sheds light on the more complicated dynamics in small mini batch regime, where estimation of a locally greedy rule may actually place the optimizer on the edge of stability of the full batch objective.

**Limitations and future directions.** We explored some limitations of the current modeling framework in Section 2.3 – in particular, the failure to capture stabilization due to higher order terms. Developing improved models (either analytically or numerically) would allow for powerful tools from other disciplines to aid algorithm design – particularly, methods from control theory. For example, state feedback schemes can be designed through the analysis of nonlinear dynamical systems to ensure asymptotic stabilization (Isidori, 1995, Chapter 7). We believe a cross disciplinary approach will be useful for designing the next generation of learning rate tuners.

The proposed CDAT rule may also help to understand and refine the design of learning rate schedules through scaling ladders (Wortsman et al., 2024). Recent work has shown that transfer of learning rates over different scales is related to consistency of curvatue dynamics (Noci et al., 2024); this suggests that approaches like ours may be useful to increase predictability of optimal learning rates across scale.

**Acknowledgements.** We thank James Martens and Mihaela Rosca for fruiftul discussions on related ideas. We also thank the reviewers for their insightful comments that helped us refine the manuscript.

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

# A  Theoretical Model

## A.1  Intuition for model of curvature dynamics

In Section 2.3, we extend the model from Damian et al. (2023) in order to understand the curvature dynamics. Here we provide some intuition for the basic structure of the model.

The analysis in Damian et al. (2023) is based on a 3rd order expansion of the loss function. A third order approximation of $f$ around some $w_0$ reads

$$f(w) \approx f(w_0) + \nabla f(w_0) \cdot (w - w_0) + \frac{1}{2}(w - w_0)^\top \nabla^2 f(w_0)(w - w_0)$$
$$+ \frac{1}{6}\nabla^3 f(w_0)[w - w_0, w - w_0, w - w_0].$$

Assuming without loss of generality that $w_0 = 0$, $f(w_0) = 0$, we can write:

$$f(w) \approx \nabla f(w_0) \cdot w + \frac{1}{2}w^\top \nabla^2 f(w_0)w + \frac{1}{6}\nabla^3 f(w_0)[w, w, w] \tag{11}$$

Consider the gradient descent dynamics with learning rate $\eta$ under this model. We can write:

$$w_{t+1} - w_t \approx -\eta(\nabla f(w_0) + H(w_t)w_t) \tag{12}$$

where $H(w_t) \equiv \nabla^2 f(w_0) + \frac{1}{2}\nabla^3 f(w_0)[w_t, \cdot, \cdot]$ is the Hessian at the current parameter $w_t$.

Let $v$ be the direction of the largest eigenvalue of $H(w_0)$. For small enough $w_t - w_0$, this is a good approximation of the largest eigendirection of $H(w_t)$. Consider the dynamics of the projection $x_t \equiv v \cdot w_t$. We make the additional assumption that $v \cdot \nabla f(w_0) = 0$ (which can be achieved with a coordinate transformation). We then get

$$x_{t+1} \approx (1 - \eta\lambda_t)x_t \tag{13}$$

where $\lambda_t := \lambda(w_t) := \lambda_{\max}H(w_t)$ is the largest eigenvalue of the current Hessian $H(w_t)$. The dynamics of $\lambda_t$ are generally slower than the dynamics of $w_t$ (and $x_t$), and are governed approximately by

$$\lambda_{t+1} - \lambda_t \approx (\nabla\lambda(w_t)) \cdot (w_{t+1} - w_t) = -\eta(\nabla\lambda(w_t)) \cdot \nabla f(w_t) \tag{14}$$

This gradient is given by

$$\nabla\lambda(w_t) = \nabla(v^\top H(w_t)v) \approx \nabla^3 f(w_0)[v, v, \cdot]. \tag{15}$$

For very small $x_t$, it has been observed (see e.g. Cohen et al. (2021)) that $\lambda_{t+1} - \lambda_t$ is increasing during most of training. Early in this regime $x_t$ is often very close to $0$ (that is, the gradient has small component in the $v$ direction). Therefore we are interested in the contribution of the gradient orthogonal to $v$, $\nabla f(w_t)_\perp \equiv (I - vv^\top)\nabla f(w_t)$, to the dynamics. We define $a_t \equiv -(\nabla\lambda(w_t) \cdot \nabla f(w_t)_\perp)$ as the instantaneous change in the top eigenvalue, due to gradient contributions orthogonal to $v$. We assume that this contribution is positive and, to simplify, independent of $t$, that is, $a_t := a > 0$.

From (13), while $\lambda_t < 2/\eta$, $x_t$ is decreasing and from (14) $\lambda_t$ is increasing.

Once $\lambda_t > 2/\eta$, the dynamics of $x_t$ changes from convergence towards $0$ to increasing values with alternating sign. We can write down the contribution to the gradient from $x_t$ using our third order expansion of the loss. We have:

$$\nabla_w f(w_0 + x_t v) \approx \nabla f(w_0) + x_t H(w_t)v + \frac{1}{2}x_t^2 \nabla^3 f(w_0)[v, v, \cdot] \tag{16}$$

We can now understand how these terms contribute to the dynamics of $\lambda_t$. The first term contributes to sharpening (increasing $\lambda_t$) via the constant $a$. The second term has oscillating sign and its long term contribution to $\lambda_t$ is small. From Equation 13 we know that

$$x_{t+1} + x_t \approx (2 - \eta\lambda_t)x_t \tag{17}$$

Near the edge of stability, $\eta\lambda_t$ is close to 2, and the sum of successive $x_t$ are small; therefore we neglect the linear $x_t$ term in Equation 16.

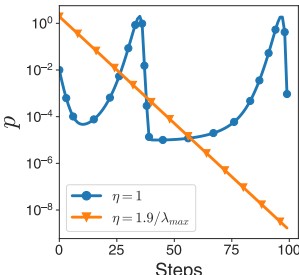
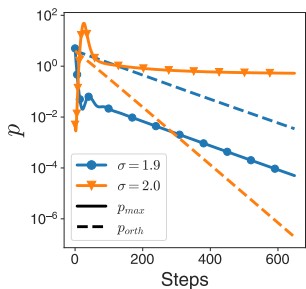

Figure 10: Dynamics of the largest eigenvalue projection $p$. For constant learning rate, $p$ cycles between stability and instability and $\lambda_{\max}$ stabilizes (blue). If learning rate tuner sets $\eta_t = 2(1-\epsilon)/\lambda_{\max,t}$, $p$ decays to 0 and there is no negative feedback preventing sharpening.

Figure 11: Dynamics of $p$ for toy model of CDAT. For $\sigma = 2$, $p$ is stable, which induces stabilization of sharpness (orange). For $\sigma = 2 - \epsilon$ for small $\epsilon$ (blue, $\epsilon = 0.1$), $p$ decreases but the ratio of $p$ to the orthogonal projection remains constant which stabilizes $y$ near 0.

The sign of the $x_t^2$ term is constant and there are no such issues. Therefore, in the non-linear regime the dynamics of $\lambda_t$ can be approximated by:

$$\lambda_{t+1} - \lambda_t \approx \eta a - \frac{\eta}{2} x_t^2 \left( \nabla^3 f(w_0)[v, v, \cdot] \cdot \nabla^3 f(w_0)[v, v, \cdot] \right) \tag{18}$$

If we define $b \equiv \frac{1}{2} \left( \nabla^3 f(w_0)[v, v, \cdot] \cdot \nabla^3 f(w_0)[v, v, \cdot] \right)$ (guaranteed to be positive), then we arrive at the dynamical equations presented in Equation 4:

$$x_{t+1} - x_t = (2 - \eta\lambda_t)x_t, \ \lambda_{t+1} - \lambda_t = \eta(a - bx_t^2) \tag{19}$$

This system leads to a quasi-stable oscillation of $\lambda_t$ around $2/\eta$; when $\lambda_t < 2/\eta$, $x_t$ goes to 0, and eventually $\lambda_t$ increases; when $\lambda_t > 2/\eta$, $x_t$ increases in magnitude, eventually leading to a decrease in $\lambda_t$.

This heuristic argument is made rigorous in the appropriate regimes in Damian et al. (2023); there they compute the difference between the full dynamics and the dynamics with the gradient projected away from the largest eigendirection using a coupling argument. The key point is that higher order contributions to the gradient are guaranteed to be anti-aligned with the gradient of the large Hessian eigenvalues. This leads to a *restoring force* which counteracts eigenvalue growth (progressive sharpening) if the largest eigenmode becomes unstable.

### A.2 Dynamics of rescaled variables

In Section 2.3, we used Equation 4 to derive equations in $p \coloneqq x^2$ and $y \coloneqq \eta\lambda - 2$. We arrived at

$$p_{t+1} = (1 + y_t)^2 p_t, \ y_{t+1} = \eta_{t+1} \left[ \eta_t \left( a - b p_t \right) \right] + \left( \frac{\eta_{t+1}}{\eta_t} \right) y_t + 2 \left[ \frac{\eta_{t+1}}{\eta_t} - 1 \right]. \tag{20}$$

The dynamical equation for $p$ is obtained by squaring the equation for $x_t$ in Equation 4. To derive the equation for $y_t$, we first derive the dynamics of $\eta_t \lambda_{t+1}$ as

$$\eta_t \lambda_{t+1} = \eta_t \left[ \eta_t \left( a - b p_t \right) \right] + \eta_t \lambda_t. \tag{21}$$

We then have

$$y_{t+1} = \frac{\eta_{t+1}}{\eta_t} [\eta_t \lambda_{t+1} - 2] + 2. \tag{22}$$

Evaluating completes Equation 5.

### A.3 Dynamics of the projection on the largest eigenvector

We present results on the dynamics of $p$ in the various models; these were omitted from the main text in order to simplify the presentation. For constant learning rate, $p$ initially decreases until EOS is crossed, after which it enters into a cycle of increase and decrease (Figure 10, blue). For our model

of linesearches, where $\eta_t = 2(1 - \epsilon)/\lambda_{\max,t}$, $p$ decays to 0 quickly and there is no mediation of sharpening (orange, $\epsilon = 0.1$).

For our model of CDAT presented in Section 3.2, $p$ stabilizes for $\sigma = 2$ (Figure 11, orange). For $\sigma < 2$, the model still predicts decay of $p$, but the ratio of $p_t$ to the orthogonal component $g_t$ remains constant (Figure 11, blue). This fixed ratio stabilizes $y$ to a value near 0.

In practice, the higher order terms in the dynamics provide additional stability, in the on-edge model, which allows $p$ to stabilize as well, see Fig. 12, Fig. 17. The key is that these terms can operate when $y$ is close to 0 for long periods of time. These results suggest that additional model development is required to understand the behavior of learning rate tuners which target the EOS.

## B  Additional Experiments

### B.1  Further analyzes of base learning rate tuners

Fig. 12 completes Fig. 3 with measures of sharpening and learning rates on the settings considered in Fig. 1. For RMSProp we considered the preconditioned Hessian following the observations done by Cohen et al. (2023) that for adaptive gradient methods such as RMSProp or Adam, the sharpness of the preconditioned Hessian, rather than the sharpness of the Hessian, defines the edge of stability. Namely, recall that RMSProp takes updates of the form

$$w_{t+1} = w_t - P_t^{-1}\nabla f(w_t), \quad \text{for} P_t = \text{diag}(\sqrt{\nu_t + \varepsilon})$$

for

$$\nu_t = (1 - \beta_2)g_t^2 + \beta_2\nu_{t-1}, \quad \nu_{-1} = 0, \ g_t = \nabla f(w_t),$$

with $\beta_2$ an exponential moving average parameter. The preconditioned Hessian takes then the form

$$\tilde{H}_t = P_t^{-1/2}\nabla^2 f(w_t)P_t^{-1/2},$$

and we report $\lambda_{\max}(\tilde{H}_t)$.

We observe similar behaviors in these regimes as in Fig. 3. Namely, the sharpness or preconditioned sharpness ever increase (2nd panels of Fig. 12), while the learning rates ever decrease (1st panels of Fig. 12). The constant learning counterpart can operate above the edge of stability while the self-tuned methods generally avoid crossing the edge of stability (3rd panels of Fig. 12).

### B.2  Analyzing additional learning rate tuners

We consider the performance of two additional classical learning rate tuners, Polyak stepsize (Berrada et al., 2020; Loizou et al., 2021; Polyak, 1964) and hyper-gradient descent (Almeida et al., 1999; Baydin et al., 2018) akin to the resilient backpropagation scheme (Riedmiller and Braun, 1992).

Briefly, Polyak stepsizes consider setting the learning rate as

$$\eta_t = \min\left\{\frac{f(w_t) - f^\star}{\|\nabla f(w_t)\|^2}, \eta_{\max}\right\}, \tag{23}$$

where $f^\star = \min_w f(w)$ is the minimum of the objective set to 0 by assuming that a neural network can overfit the data, and $\eta_{\max}$ is a maximal stepsize selected as 1 or 100 in our experiments (we take the best instance).

Hyper-gradient descent considers updating the stepsize towards maximal decrease of the objective. Namely, defining the objective obtained after one step $h_t(\eta) = f(w_t + \eta u_t)$, the algorithm updates $\eta_t$ by a gradient step on $h_t$ resulting a priori in $\eta_{t+1} = \eta_t - \alpha\nabla f(w_t + \eta_t u_t)^\top u_t$ for a given hyper-learning rate $\alpha$. Almeida et al. (1999); Baydin et al. (2018) argued for using multiplicative updates of the form

$$\eta_{t+1} = \eta_t\left(1 - \beta\frac{\nabla f(w_t + \eta_t u_t)^\top u_t}{\|\nabla f(w_t + \eta_t u_t)\|_2\|u_t\|_2}\right). \tag{24}$$

Intuitively, the learning rate increases if the update is aligned with the negative gradient direction and decreases otherwise. Resilient backpropagation (Riedmiller and Braun, 1992) adopts a similar logic

componentwise. In our experiments we vary $\beta$ and select the best instance, see Appendix C.5 for more details.

We observe that Polyak stepsizes (top figure of Fig. 13) generally select larger learning rates than the constant learning rate counterpart. The efficiency of Polyak stepsizes is not reached by the CDAT rule with $\sigma = 2$ but with a slightly larger scale $\sigma = 2.06$. The efficiency of the Polyak stepsize method, in particular compared to a simple linesearch, in a full batch regime, has also been reported by Roulet et al. (2023). The proposed CDAT rule may capture the benefits of aggressive learning rates taken by Polyak stepsizes in a smoother way by allowing various scales.

On the other hand, the hyper-gradient descent performs just on par with the fine-tuned constant learning rate counterpart (bottom figure of Fig. 13). We also observe a slow, yet steady, progressive sharpening when using the hyper-gradient descent. As with the linesearch method or the quadratically greedy rule, the hyper-gradient descent focuses on selecting a learning rate that decreases the loss, which appears, across those tuners, to potentially suppress effective stabilization effects naturally appearing with constant learning rate.

### B.3 Base learning rate tuners in a stochastic regime

In Fig. 14, we report the performance of classical learning rate tuners (linesearch or quadratically greedy method) in a stochastic regime for varying batch-sizes. As observed previously by Vaswani et al. (2019) or Roulet et al. (2023), a linesearch for example can perform well in a stochastic regime. Note that the two approaches (linesearch and quadratically greedy method) display similar behaviors (just as they displayed similar behaviors in the full batch regime). This hints that, rather than playing with the numerous hyperparameters of a linesearch we may focus simply on an additional scaling factor for the quadratically greedy rule, which motivated the proposed CDAT rule.

### B.4 Targeting the edge of stability using the exact sharpness

Cohen et al. (2021, Appendix F) reported bad performance of adaptive learning rate tuners selecting the stepsize as
$$\eta = 2/\lambda_{\max}(\nabla^2 f(w)),$$
which may fix the learning rate just at the edge of stability. Note that such a definition does not take into account the additional alignment of the update with the largest eigenvector. Our proposed diagnostic rule CDAT rather considers the edge of stability given by a local approximation of the objective along the update so to take into account the alignment of the update with the largest eigenvector of the Hessian. We ran experiments with a rule
$$\eta = \sigma/\lambda_{\max}(\nabla^2 f(w)), \tag{25}$$
that lets the scaling factor vary just as done with CDAT. The only difference is in the estimation of the base estimate of the edge of stability (CDAT does it with the help of a quadratic approximation of the objective, while the rule (25) uses an exact computation of the sharpness). In Fig. 15, we observe that setting the scale $\sigma \approx 2$ leads to poor performance as previously observed by Cohen et al. (2021). Note however that by setting the scaling much above 2 (like $\sigma = 3$) such a rule may outperform a constant learning rate. This hints that the rule (25) misses the alignment of the update with the largest eigenvector, which motivated the CDAT rule.

### B.5 Analyzing instantaneous gains versus long-term gains

In Fig. 16, we investigate the difference of instantaneous decrease using the quadratically greedy rule (CDAT with $\sigma = 1$) compared to the on edge rule (CDAT with $\sigma = 2$). Throughout a training, the quadratically rule ensures a larger instantaneous decrease as intended through its definition as a learning rate that minimizes the loss. Yet, in the long term, the quadratically greedy rule underperforms the on edge rule (Fig. 5).

### B.6 Additional metrics for the CDAT rule

In Fig. 17, we additionally measure the alignment of the updates with the largest eigenvector and the angle between successive updates. We observe that the CDAT rule for $\sigma \approx 2$ behaves similarly as the constant learning rate counterpart. In particular, the updates tend to quickly be in opposed directions. The quadratically greedy rule does not demonstrate such a behavior.

### B.7 Sensibility analysis to architecture hyperparameters

In Fig. 18, we study CDAT for simple MLPs in a full batch regime on the MNIST dataset. Our goal is to understand the benefits of the proposed CDAT rule for varying hyperparameters. First, we analyze the sensibility to width and depth of an MLP in a similar fashion as Cohen et al. (2021, Appendix D) did to analyze progressive sharpening.

We observe that accrued gains can be obtained with the CDAT rule for larger widths (top left panel of Fig. 18). Note that Cohen et al. (2021, Appendix D) found less sharpening at higher widths. In terms of depth (top right panel of Fig. 18), the CDAT rule works best with larger depths while we note a slight shift of optimal scaling factors from 2 to slightly below 2.

The CDAT rule appears to work best with small or no weight decay (bottom left panel of Fig. 18) while its benefits fade with larger weight decay (no difference between greedy $\sigma = 1$ and on edge $\sigma = 2$). Finally, while the method naturally finds smaller train losses with larger subsets of data, we do not observe a significant shift of relative performance between scales as the size of the data increases (bottom right panel of Fig. 18).

### B.8 CDAT rule versus prefixed schedule in full batch regime

In Fig. 19, we compare the proposed CDAT rule with prefixed schedules in a full batch regime. We observe that while placing the optimizer on edge could improve on constant learning rate counterparts, prefixed schedules can outperform the CDAT rule. This points out that the feedback loop exploited by CDAT may miss some additional nonlinear effects that could further enhance self-tuning rules.

### B.9 Detailed performances of CDAT in stochastic regime

In Fig. 20 and Fig. 21, we detail the performances of the CDAT rule in the stochastic regime for varying batch sizes. In the stochastic regime, recall that the heatmap of the performance of CDAT in terms of batch-size heavily depended on the appropriate scaling factor Fig. 7 (in comparison a scaling factor of approximately $\sigma = 2$ appeared generally good in the full batch regime). In both Fig. 20 and Fig. 21, we observe that the method may generally work better at larger batch sizes. Understanding better the right statistics to estimate as well as appropriate estimators of the edge of stability in a stochastic regime is a future direction.

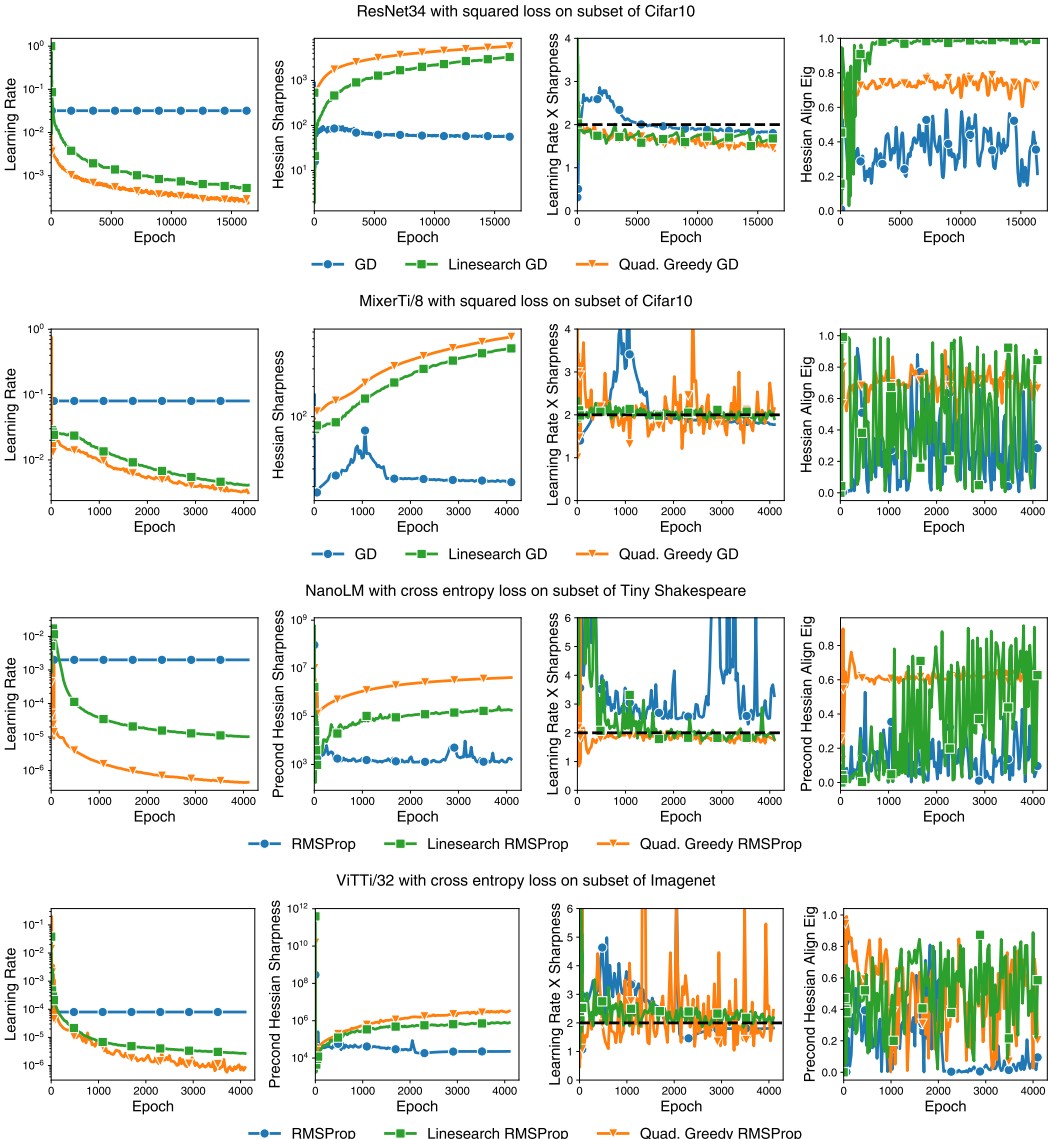

Figure 12: **Learning rates and sharpness dynamics of baseline learning rate tuners.** Learning rate, Hessian or preconditioned Hessian sharpness, their product, and the alignment between the update and the largest eigenvector of the Hessian or the preconditioned Hessian. As in Fig. 12, we observe that a linesearch (1) or a quadratically greedy (3) learning rate tuner display decreasing learning rates along training. The sharpness of the Hessian (for GD) or preconditioned Hessian (for RMSProp) keep increasing for the self-tuned baselines while they stabilize for the constant learning rate counterparts. The self-tuned methods perform generally below the edge of stability or at least much less above than the constant learning rate counterpart.

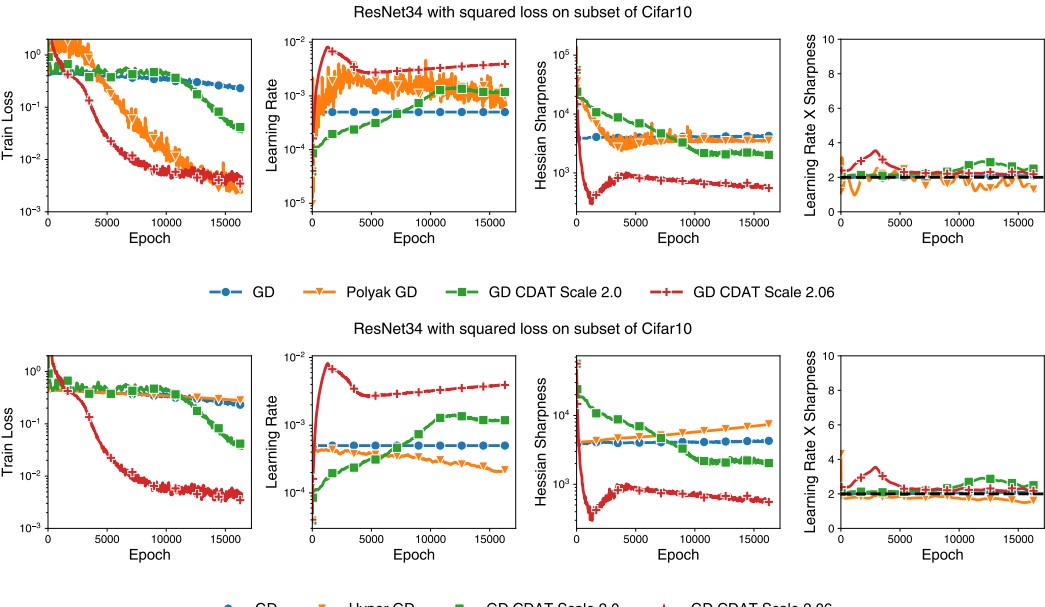

Figure 13: **Analyzing additional learning rate tuners.** Train loss, learning rate, sharpness, and their product along training with gradient descent using a constant or self-tuned learning rate with various tuners. Polyak stepizes (23) are effective in a full batch regime (top figure) outperforming CDAT on edge ($\sigma = 2$). The effectiveness of the Polyak stepsizes are partially captured by an aggressive CDAT rule placing the optimizer on edge $\sigma = 2.06$. On the other hand, a hyper-gradient descent (24) performs just on par with the constant learning rate counterpart in this regime. It also displays an ever-increasing sharpening akin to the one observed for a linesearch or the quadratically greedy rule Fig. 3.

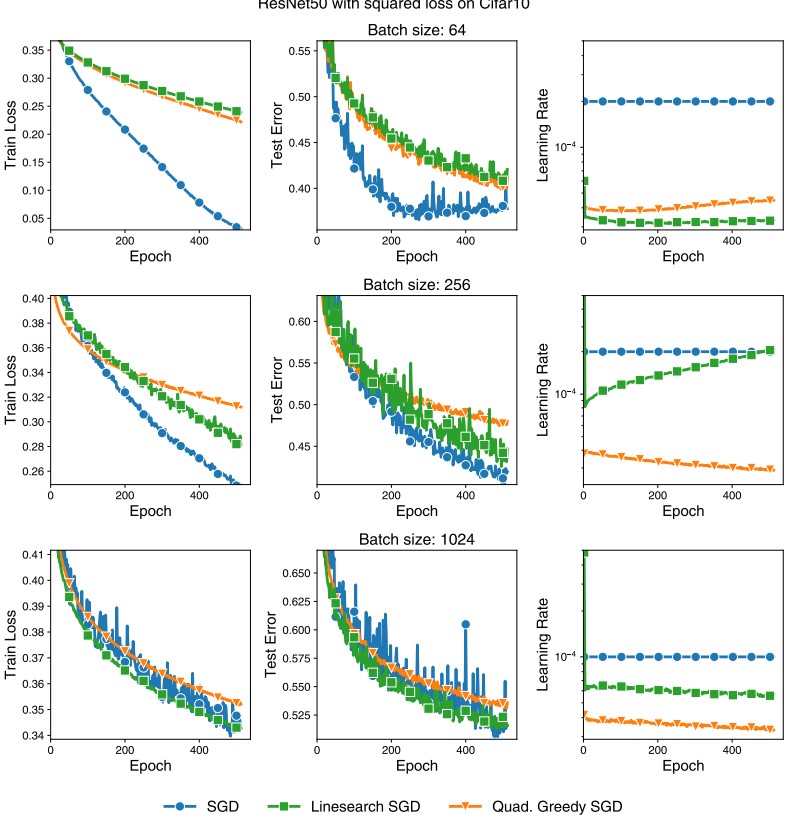

Figure 14: **Classical learning rate tuners can work well in stochastic regime.** In a stochastic regime, we observe that, e.g., a linesearch can perform on par or even better than the constant learning counterparts. However, this good performance is not explained by the common belief that linesearches work well in a full batch regime (Fig. 1). The linesearch and quadratically greedy rule perform similarly in this setting.

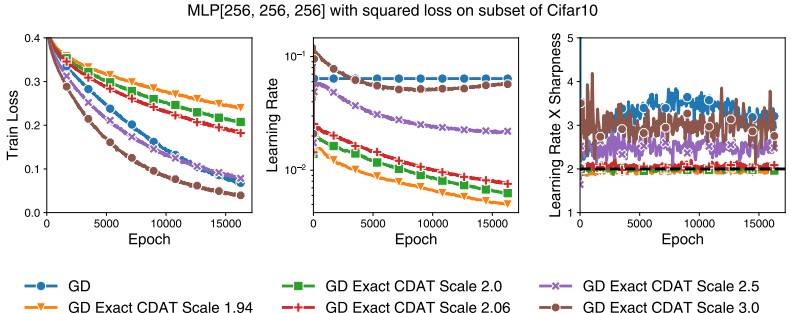

Figure 15: **On edge with exact sharpness.** We consider a rule similar to CDAT (8) but using the exact sharpness, that is the largest eigenvalue of the Hessian, as a base learning rate while varying an additional scale factor (see (25)). By using the exact sharpness, a scaling factor of $\sigma = 2$ leads now to poor performance, while a scaling factor of $\sigma = 3$ is performant. By using the exact sharpness (25) we do not take into account the actual alignment of the update with the largest eigenvector which may explain the shift of optimal scaling factors in this case.

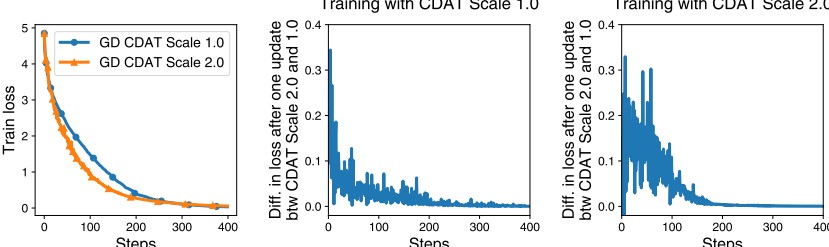

Figure 16: **The quadratically greedy rule ensures larger instantaneous decrease of the loss.**
Train losses and difference in loss between one step using the on-edge rule (CDAT, scale=2) and
one step using quadratically greedy rule (CDAT, scale=1). Results averaged over 10 initializations
and disjoint 4096-sample subsets of CIFAR100. MLP architecture: single hidden layer of size
1024, ReLU activations, trained with GD and cross-entropy loss in a full batch setting. We plot
$f(w_t + \eta^{\text{oe}} u_t) - f(w_t + \eta^{\text{qg}} u_t)$ along a training performed either with the quadratically greedy rule
($\sigma = 1$) or the on-edge rule ($\sigma = 2$). In both cases, this difference is positive meaning that the
quadratically greedy rule ensures a larger instantaneous decrease of the loss. Yet the quadratically
greedy rule underperforms in the long term (see Fig. 5, also holds in this full batch setting).

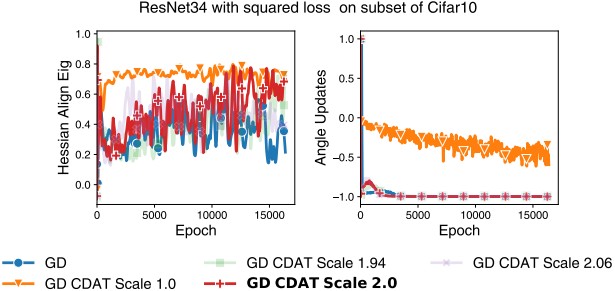

Figure 17: **Further analysis of the CDAT rule.** This is the same setting as in Fig. 6 except that
we plot the alignment between the largest eigenvector of the Hessian and the update, and the angle
between successive updates. We observe that the CDAT rule for $\sigma \approx 2$ behaves similarly as the
constant learning rate counterpart. In particular, the updates tend to quickly be in opposed directions.
The quadratically greedy rule does not demonstrate such a behavior.

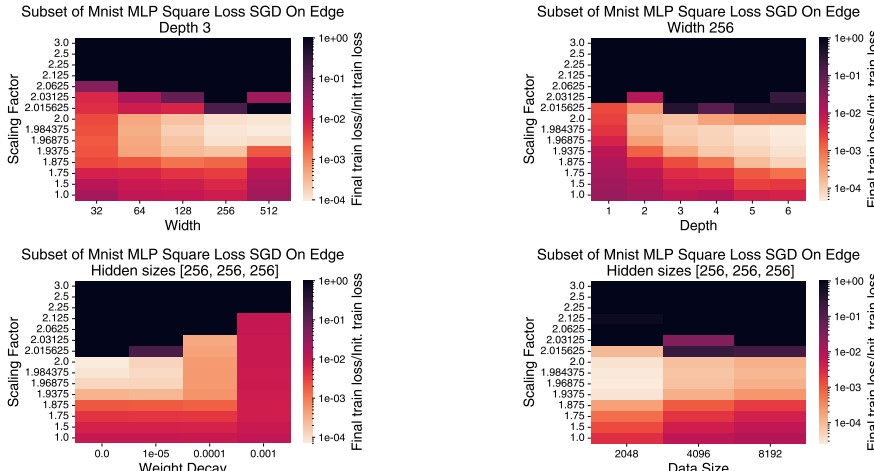

Figure 18: **Improvements of CDAT rule on edge for varying hyperparameters.** Varying width, depth, weight decay and size of the subset considered when using the CDAT rule with varying scaling factors. We observe that accrued gains can be obtained with the CDAT rule for larger widths (top left panel). In terms of depth (top right panel), the CDAT rule works best with larger depths, while we note there a slight shift of optimal scaling factors from 2 to slightly below 2. The CDAT rule appears to work best with small or no weight decay (bottom left panel) while its benefits fade with larger weight decay (no difference between greedy $\sigma = 1$ and on edge $\sigma = 2$). Finally, while the method naturally finds smaller train losses with larger subsets of data, we do not observe a significant shift of relative performance between scales as the size of the data increases (bottom right panel).

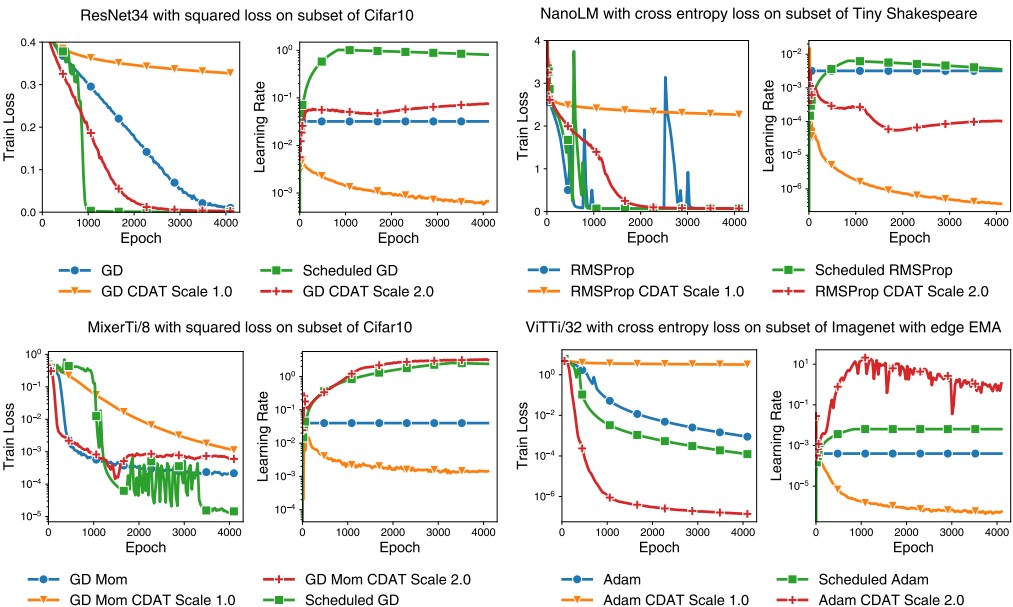

Figure 19: **CDAT rule may not fully capture the benefits of pre-fixed schedules**. Train loss and learning rate behaviors for fine-tuned optimizers with or without schedules vs self-tuned counterparts with CDAT on various architecture, datasets, losses in a full batch regime. While the CDAT rule displays a behavior to warmup schedules, it does not completely catch the benefits of pre-fixed schedules. Note for the top-left part that the performance of the pre-fixed schedule is akin to the performance reported with CDAT $\sigma = 2.5$ at early times suggesting that a varying scaling factor, or taking higher order dynamics may be important to fully capture the benefits of warm-up schedules.

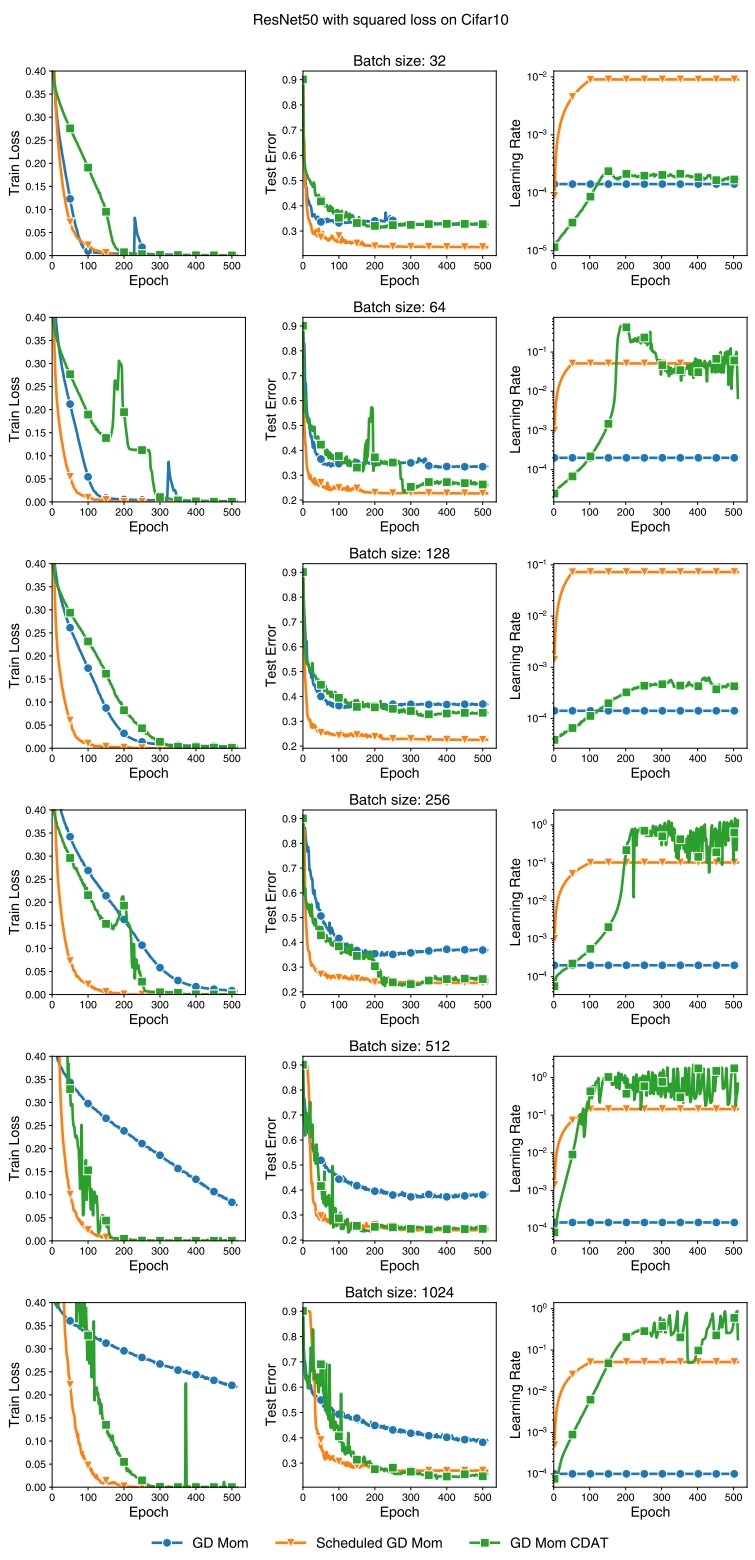

Figure 20: **Performance of fine-tuned algorithms in stochastic regime with SGD Momentum.** In the stochastic regime with SGD momentum, we observe that the CDAT rule may outperform the constant learning rate counterpart (particularly for large batch sizes) while performing on par or underperforming the scheduled learning rate counterparts. Interestingly, a warm-up phase appears naturally induced by the CDAT rule (right plots).

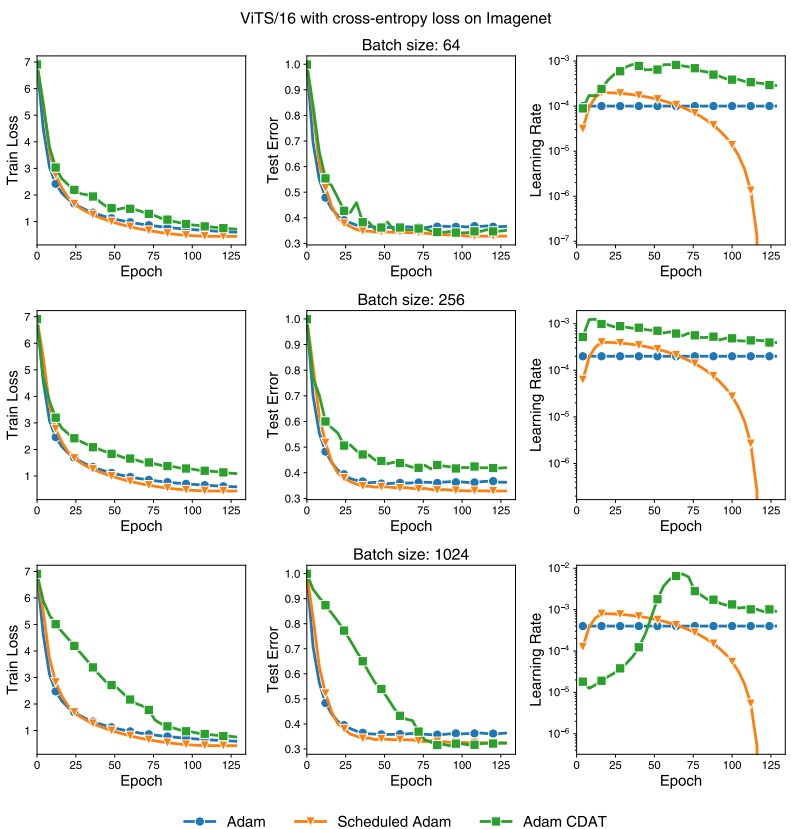

Figure 21: **Performance of fine-tuned algorithms in stochastic regime with Adam.** In the stochastic regime with Adam we observe varying performance of the CDAT rule, generally on par or underperforming the baselines. Several factors can explain the underperformance. The scaling factor is not well understood, and a finer grid search could improve the scheme. Similarly, a better estimate or a better understanding of the edge of stability in a stochastic regime could improve the approach. Tackling curvature dynamics by analyzing feedback effects as in the CDAT rule may help such design.

# C  Experimental Details

## C.1  Datasets

**MNIST.**   MNIST is an image classification dataset of handwritten digits (LeCun et al., 2010). We normalized the pictures so that each pixel is between 0 and 1. We did not standardize the data. We only used this dataset to test varying MLP architectures in Fig. 18. See Appendix C.5 for any additional relevant details.

**CIFAR10.**   CIFAR10 is an image classification dataset of colored images of size $32 \times 32$ with 10 classes (Krizhevsky et al., 2009). We normalized the pictures so that each pixel is between 0 and 1. We did not standardize the data. In the full batch regime, we considered a subset of 4096 samples. In the mini batch regime, we considered the full dataset of $50,000$ samples for training (dropping out the remainder batch) and tested on the $10,000$ validation samples.

**Tiny Shakespeare.**   This consists in $40,000$ lines of Shakespeare from a variety of Shakespeare's plays (Karpathy, 2015). The task consists in a character-level prediction task among 64 characters. In the full batch regime, we considered a subset of 2048 samples consisting of blocks of 64 characters.

**Imagenet.**   Imagenet is an image classification dataset of various images from the web (Deng et al., 2009). The images have various sizes. The original Imagenet-1K dataset contains 1000 classes. For the full batch experiments, we consider the Imagenette (Howard, 2019) subset that consists in only 10 classes and took 1024 samples out of it. We consider the usual prepreocessing for Imagenet as detailed in the Scenic library (Dehghani et al., 2022). Namely, for training we consider random cropping and random flip at training. For testing, we center crop the images. Each time the cropping reduces the colored images to a $224 \times 224$ size. In the mini-batch regime we consider the complete training dataset of 1.2 million images (Imagenet-1K), dropped the remainder batch, and reported test error on the $50,000$ validation images.

## C.2  Architectures

**Residual Network (ResNet).**   We considered the standard ResNet architectures (ResNet34, ResNet50) of He et al. (2016) as implemented in the Scenic library (Dehghani et al., 2022). For the examples with ResNet34 we removed the batch normalization layers (Ioffe and Szegedy, 2015). For the examples with ResNet50 we replace the batch normalization layers with layer normalization layers (Ba et al., 2016).

**Multi-Layer Perceptron (MLP) Mixer.**   We consider the standard MLP Mixer architectures (Tolstikhin et al., 2021) as implemented in the Scenic library (Dehghani et al., 2022). By Mixer Ti/8, we mean the tiny model of Mixer provided in the Scenic library (see `https://github.com/google-research/scenic/blob/main/scenic/projects/baselines/configs/imagenet/imagenet_augreg_mixer_config.py`) with patches of size $8 \times 8$. We removed dropout (both layer and depth wise).

**Nano Language Model (NanoLM).**   We consider a simpel sequence-to-sequence Transformer model implemented in Optax (`https://github.com/google-deepmind/optax/blob/main/examples/nanolm.ipynb`). The model consists of 6 stacked transformer blocks, each of which contains a multi-head attention layer followed by a feed-forward layer. Layer normalization is used used within the transformer blocks to improve training stability. Finally, a dense layer maps the model's output to the vocabulary size, producing probabilities for each character as the next potential character. The only difference with respect to the previous code is that we removed dropout (for deterministic training) and, as mentioned in the previous subsection, reduced the size of the dataset to be able to estimate the sharpness.

**Vision Transformer (ViT).**   We consider the standard Vision Transformer architecture (Dosovitskiy et al., 2021) as implemented in the Scenic library (Dehghani et al., 2022). By ViT Ti/32, we mean the tiny model of Vision Transformer provided in the Scenic library (see `https://github.com/google-research/scenic/blob/main/scenic/projects/`

`baselines/configs/imagenet/imagenet_augreg_vit_config.py`) with patches of size $32 \times 32$. We removed dropout (both layer and depth wise).

**Weight decay.** In all examples, except mentioned otherwise, we consider a fixed weight decay of $10^{-5}$. See Fig. 18 (bottom left panel) for an analysis of sensitivity of the proposed CDAT rule with varying weight decay.

### C.3 Algorithms

In all experiments, when selecting the "best" tuner we considered the average train loss over the last 5 iterates.

**Fine-tuning base optimizers.** The implementation of all optimizers are taken from the Optax library (DeepMind et al., 2020). In all experiments, we fix all hyperparameters of the base optimizer ((S)GD, (S)GD with Momentum, RMSProp, Adam) to their default values: $0.9$ for the momentum of (S)GD with Momentum, $0.999$ for the EMA parameter of the second moment of RMSProp and Adam, $0.9$ for the EMA parameter of the first moment of Adam. We fine-tune the learning rate on a logarithmic base of 2 around a base learning rate such as $10^{-3}$ or $10^{-4}$ (depending on the algorithm, the architecture and the mini-batch size in the stochastic regime as detailed below in Appendix C.5), while making sure that the grid is sufficiently large such that the best learning rate is found inside the grid and not as the smallest or the largest.

For the scheduled versions of the base optimizers, we consider three shapes: linear warm-up followed by constant learning rate, linear warm-up followed by linear decay, linear warm-up followed by cosine decay. The number of iterations for the warm-up period is chosen as a fraction of the overall number of steps detailed in Appendix C.5. We also varied the horizon for the decaying schedules, see again Appendix C.5.

**Implementation of the linesearch procedure.** To implement the linesearch procedure described in Section 2, we consider the following criterion

$$f(w_t + \eta_t^{\mathrm{ls}} u_t) \leq (1 + \delta) f(w_t) + c \eta_t^{\mathrm{ls}} \nabla f(w_t)^\top u_t.$$

Compared to (1), we added a relative decrease hyperparmeter $\delta$ as we observed that the linesearch can sometimes stay stuck at vanishing learning rates otherwise.

To find a valid criterion we consider a usual backtracking linesearch that starts from a guess $\eta_{t,0} = \min\{c_+ \eta_{t-1}^{\mathrm{ls}}, 1\}$. Choosing $c_+ = +\infty$ means that we start with an initial guess of 1 at each iteration. The learning rate is then decreased by a factor $c_-$ until the criterion is satisfied. Formally, the selected stepsize is then

$$\eta_t^{\mathrm{ls}} = \max\{\eta_{t,k} = c_-^k \eta_{t,0} : f(w_t + \eta_{t,k} u_t) \leq (1 + \delta) f(w_t) + c \eta_{t,k} \nabla f(w_t)^\top u_t\}.$$

We run the search until the criterion is satisfied in the full batch regime and for a maximum of 30 iterations in the mini-batch regime. In the experiments, we consider the following variations.

- $c \in [0, 10^{-4}, 0.5]$,
- $c_+ \in [4, +\infty]$,
- $c_- \in [0.8, 0.9]$,
- $\delta \in [0, 1e-3]$ for $c = 0$ and $\delta = 0$, for $c \in [10^{-4}, 0.5]$.

**Implementation of quadratically greedy tuner and CDAT.** To implement the quadratically greedy tuner or CDAT, we compute the denominator $u^\top \nabla^2 f(w) u$ as the second partial derivative of $f$ along $u$, that is,

$$u^\top \nabla^2 f(w) u = \partial^2 f(w)[u, u] = \partial(\partial f(\cdot)[u])(w)[u],$$

where $\partial g(w)[u]$ amounts to a Jacobian vector product (jvp) computed with forward mode auto-diff in differentiable programming languages such as JAX (DeepMind et al., 2020).

Computing the denominator in the CDAT rule by forward mode automatic differentiation enables a much lower memory consumption than using Hessian vector products (see, e.g., (Blondel and Roulet, 2024, Chapter 8), (Dagréou et al., 2024) for more details). The computation of the denominator by applying twice forward mode automatic differentiation still incurs approximately three times the

memory necessary to compute the objective (Blondel and Roulet, 2024, Chapter 8). The computational cost of computing the denominator is also approximately three times the computational cost of computing the objective. The above approximations are done with the following reasoning. The second partial derivative requires to follow the graph of computation but with three variables, one for the parameters, one for a copy of the update direction, one for another copy. At each node in the computation graph, the program computes the original computation, computes its first derivative along the first copy of the update direction, and computes the second derivative along the second copy.

In practice, we observed for, e.g., the experiment on the full Imagenet dataset in mini-batch that the proposed CDAT rule required twice the wall time of the constant or scheduled learning rates counterparts. For this project, we considered CDAT as a diagnostic tool to understand the interplay between curvature and learning rate tuners. For future work, the cost of computing the approximate edge may be circumvented or amortized by using, e.g., parabolic approximations as done by Mutschler and Zell (2020), or by computing it at given intervals as done by Liu et al. (2024).

**Further justification for the CDAT formula.**    In (8) we took the absolute value of the denominator to deal with concave approximations. Eigenvalue modifications in a Newton method are discussed by Nocedal and Wright (1999, Section 3.4). Taking the absolute value is one possible option. In practice, we observed positive curvatures along the update direction such that this choice did not matter.

### C.4   Metrics implementation

**Sharpness estimation.**    We estimated the sharpness by a power iteration method run for $1000$ iterations with an early stopping criterion defined by less than $10^{-3}$ relative accuracy. We accessed the Hessian by Hessian vector products, which limited the size of the full batch datasets considered on TPUs. The power iteration a priori returns the largest eigenvalue in magnitude $|\lambda|_{\max}$ and not necessarily the largest positive eigenvalue $\lambda_{\max}$. But in practice the largest eigenvalue in magnitude is the largest eigenvalue, see, e.g., (Ghorbani et al., 2019) for an in-depth study of the spectrum of the Hessian along the iterations of deep learning.

### C.5   Additional experimental details per figure

We detail here any additional detail per figure not detailed in the summary above.

**Fig. 1.**    The ResNet34 has no batch normalization layers. For the GD baseline on ResNet and the MLP Mixer the constant learning rate was tuned on a grid $\{10^{-3} \cdot 2^i, i \in \{-1, \ldots, 7\}\}$. For the RMSPRop baseline on the NanoLM and ViT, the constant learning rate was tuned on a grid $\{10^{-3} \cdot 2^i, i \in \{-1, \ldots 7\}\}$ and $\{10^{-5} \cdot 2^i, i \in \{0, \ldots 7\}\}$ respectively.

**Fig. 2.**    We consider a linear classification (in other words using an MLP without hidden layers) on the subset of CIFAR10 detailed above. We search the constant stepsize of gradient descent in $\{10^{-3} \cdot 2^i, i \in \{0, 1, 2\}]\}$. The grid is centered around $1/\|\nabla^2 f(w_0)\|_2$, that is the optimal stepsize in a convex smooth setting. As for above experiments, the largest learning rate in the grid led to divergence.

**Fig. 3.**    This is the same setting as in the first panel of Fig. 1.

**Fig. 4.**    For constant learning rate, model settings were given by $a = 3 \cdot 10^{-2}$, $b = 3 \cdot 10^{-1}$, $\eta = 1$. For fixed $y = -0.1$ training, $a = 1$, $b = 0.5$, $\eta_0 = 1.0$.

**Fig. 5.**    We considered the same settings as in Fig. 1. The grid search for GD on ResNet34 and RMSProp on NanoLM are the same as in Fig. 1. For the MLPMixer, we fine-tuned GD with momentum on a grid $\{10^{-2} \cdot 2^i : i \in \{1, \ldots, 8\}\}$. For the ViT, we fine-tuned Adam on a grid $\{10^{-5} \cdot 2^i : i \in \{1, \ldots, 8\}\}$.

**Fig. 6.**    This is exactly the same setting as in the first panel of Fig. 5.

**Fig. 7.** We considered the full dataset of CIFAR10 with ResNet50 with layer normalization in place of batch normalization.

**Fig. 8.** See the details given for Fig. 20 and Fig. 21.

**Fig. 9.** For both values of $\sigma$, $a = 5 \cdot 10^{-2}$, $b = 10^{-1}$, $\nu = 0.1$, $\lambda_0 = 18$, $\eta_0 = 0.05$, $g_0 = p_0 = 4$.

**Fig. 10.** Same settings as Figure Fig. 4.

**Fig. 11.** Same settings as Figure Fig. 9.

**Fig. 12.** This is the same setting as in Fig. 1.

**Fig. 13.** We consider the same setting as in Fig. 3. For the Polyak stepsizes (23), we let $\eta_{\max}$ vary between 1 and 100 and select the best. For the hypergradient descent, we let the hyper learning rate vary in $\beta \in \{10^i, i \in \{-3, \ldots, 0\}\}$.

**Fig. 14.** We considered again the ResNet50 with layer normalization instead of batch normalization. For the constant learning rate baseline we searched over a grid of $\{\eta_m \cdot 2^i, i \in \{-1, \ldots, 5\}]\}$, for $\eta_m = 10^{-4} \cdot \sqrt{m/4096}$ for $m$ the batch size.

**Fig. 15.** We considered a simple MLP with hidden sizes $(256, 256, 256)$, ReLU activations. We tuned the constant learning rate baseline on $\{10^{-3} \cdot 2^i, i \in \{-1, 7\}]\}$.

**Fig. 16.** Details are provided in the legend.

**Fig. 17.** This is the same setting as in Fig. 6.

**Fig. 18.** In this figure, the MLPs considered use ReLu activations. If not detailed, the weight decay is set to $10^{-5}$ and the subset considered is of size 8192.

**Fig. 19.** The settings are the same as in Fig. 5. For the constant learning rate baselines we searched on a gird $\{\eta_{\text{base}} \cdot 2^i, i \in \{-3, \ldots, 9\}\}$. The base learning rate $\eta_{\text{base}}$ was chosen to be $10^{-3}$ for ResNet, $10^{-2}$ for the Mixer, $10^{-4}$ for the NanoLM and ViT. For the schedules' shapes, we searched over linear warm-up, linear warm-up with linear decay, linear warm-up with cosine decay. The initial and end learning rate were set to 0. The horizons for the schedules were chosen in $[N, N/2, N/4]$ for $N = 8192$ for the NanoLM, $N = 16384$ for the ViT, Mixer and ResNet. The fraction of warm-up steps was searched in $\{0.05, 0.1, 0.2\}$.

**Fig. 20.** For the constant learning rate baseline, we consider searching the best constant learning rate on a grid $\{\eta_m \cdot 2^i, i \in \{-1, \ldots, 7\}\}$ for $\eta_m = 10^{-4} \cdot \sqrt{m/4096}$ where $m$ denotes the varying batch size.

For the scheduled baseline, we consider the variants presented above (linear warm-up followed by constant, linear warm-up followed by cosine decay, linear warm-up followed by linear decay) with varying fraction of warm-up steps $(0.05, 0.1, 0.2)$ and an initial learning rate of 0, a final learning rate of 0 for a fixed horizon of 512 epochs, and a peak learning rate searched over $\{\eta_m \cdot 4^i, i \in \{4, \ldots, 9\}\}$.

The scaling factor $\sigma$ of CDAT was searched on a grid $\{0.4, 0.6, \ldots, 2.8\}$, and we also tuned the EMA parameter $\beta_{\text{cdat}}$ in the computation of the numerators and denominators of the edge in $\{0, 0.9, 0.99\}$. The best parameters found for CDAT can be inferred from Fig. 7. Namely, we found that non-zero EMA parameter for the estimation of the edge decay was essential for good performance and that the best scaling factor varied with the batch size. For example, at batch size 256 the best scaling factor is $\sigma = 1.8$ with $\beta_{\text{cdat}} = 0.9$.

**Fig. 21.** For the constant learning rate baseline, we consider searching the best constant learning rate on a grid $\{\eta_m \cdot 2^i, i \in \{-1, \ldots, 7\}\}$ for $\eta_m = 10^{-4} \cdot \sqrt{m/1024}$ where $m$ denotes the varying batch size.

For the scheduled baseline, we consider a linear warm-up followed by cosine decay, with a fraction of warm-up steps of $0.1$ and an initial learning rate of 0, a final learning rate of 0 for a fixed horizon of 128 epochs, and a peak learning rate searched over $\{\eta_m \cdot 4^i, i \in \{1, \ldots, 5\}\}$.

The scaling factor $\sigma$ of CDAT was searched on a grid $\{0.4, 0.6, \ldots, 2.6\}$, and we also tuned the EMA parameter $\beta_{\text{cdat}}$ in the computation of the numerators and denominators of the edge in $\{0, 0.9, 0.99\}$.

### C.6  Assets license and computing ressources

**Assets.**  All experiments are done in the open-source JAX ecosystem (DeepMind et al., 2020): architectures are taken from Scenic (Dehghani et al., 2022), datasets from TensorFlow Dataset, algorithms from Optax. The datasets are MNIST (LeCun et al., 2010), (Creative Commons Attribution-Share Alike 3.0 license) CIFAR10 (Krizhevsky et al., 2009) (no available license), Imagenet (Deng et al., 2009) (ImageNet explicitly permits the use of the dataset for non-commercial research purposes, however there is no single license since the images are scrapped from different sources with different licenses), TinyShakespeare (Karpathy, 2015) (Apache 2.0 license in TensorFlow dataset, though the works of William Shakespeare are in the public domain).

**Computing resources.**  Experiments have mostly been run on Tensor Processing Units (TPUs) v2 (180 Tera Floating-Point Operations per Second (TFLOPS), 64 GB High Bandwidth Memory (HBM)). Experiments on MLP Mixers required TPUs v3 (420 TFLOPS 128 GB HBM). Very small scale experiments on MNIST with MLPs were run on CPUs. In terms of wall time, as discussed in Appendix C.3, we observed that the CDAT rule can be twice slower than the constant or scheduled learning rate counterparts. We consider CDAT as a diagnostic tool and leave as future work efficient implementations. Preliminary experiments and additional attempts to further adapt the momentum parameter on edge are not reported.

**Authors contributions.**

- Vincent Roulet conducted the experimental work from the failures of linesearches to the analysis of the CDAT rule in various settings.
- Atish Agarwala developed the theoretical model, with associated figures, interpretations and comments. He also helped to guide the experimental study with the insights gathered by the model.
- Jean Bastien Grill did an initial empirical study that gathered first intuitions on the method. He participated in the discussions and contributed to the writing.
- Grzegorz Swirszcz participated in the discussions.
- Mathieu Blondel participated in the discussions, contributed to the writing and proposed an alternative rule using a Gauss-Newton approximation of the objective.
- Fabian Pedregosa initiated the project, performed a larger scale empirical study of the CDAT rule on the MLCommons benchmark, participated in the discussions, and contributed to the writing.

