# OpenReview forum: "Stepping on the Edge: Curvature Aware Learning Rate Tuners"
_NeurIPS.cc/2024/Conference — NeurIPS 2024 poster_

### Official Review · Reviewer_jCtW · 2024-06-25

**Soundness:** 2
**Presentation:** 2
**Contribution:** 2
**Rating:** 5
**Confidence:** 3

**Summary:**

This paper intents to design new learning rate tuners based on insights from the Edge of Stability phenomenon. Based on the conjecture that classsical linesearch methods undershoot the edge of stability, and that this causes poor performance, the paper proposes a new learning rate tuner, that targets ceratin sharpness levels.

**Strengths:**

Learning rate tuning is tedious and time/compute consuming, and classical linesearch techniques often underperform. Therefore, this paper adresses an important problem for machine learning practicioners. The connection between Edge of Stability dynamics, and LR scheduling looks novel to me.
The proposed method is compatible with any update direction, thus directly applicable to Adam or similar optimizers.

**Weaknesses:**

The proposed CDAT tuner has several limitations:

1) It requires additional computation, such as  vector-Hessian-vector products. How much does CDAT increase the runtime for one step?
2) The performance looks highly sensitive to the choice of $\sigma$ (see Fig. 5). In particular, choosing $\sigma=2.0$ does not always beat tuned GD, even for rather small and easy datasets (subset of CIFAR10). It seems that CDAT replaces the necessity to tune the learning rate with tuning $\sigma$.
3) Only few experiments are conducted in a stochastic regime, which is the one that is interesting for practical use, and the results are not very convincing (e.g. Fig. 8 bottom).

Besides those questions whether CDAT can be effective for tuning the learning rate, I have the following doubt regarding its motivation:

In Figure 3, Gradient Descent (GD) is also below the EoS for much of the training (third plot, from epoch 10000 onwards, and similar in Fig 12 for MixerTi/8). The paper conjectures that the poor performance of LR tuners comes from the fact that they stay below EoS systematically (Fig. 4 and end of section 2). However, from the observation above, it seems to me that the same reasoning would apply to GD with constant learning rate for some experiments. Thus, the main motivation for the derivation of CDAT, namely to target EoS, seems not fully convincing in the first place. In case I missed something important here, I am happy to discuss further.

**Questions:**

* Why do you use the squared loss for the CIFAR dataset, instead of the cross-entropy loss?
* In the Damian et al. paper (ref. [34]), the dynamics are not the same as what you write in (4): in particular, they define quantity $y_t$ where you write $\lambda_t$, and those are not the same in my understanding. Can you clarify? (The quantity $y_t$ in [34] is roughly sharpness - 2/lr)
* Why is Fig. 4 using model (4), and not model (5), if the goal is to illustrate learning rate tuners, where the LR depends also of time (model (5))?
* Fig. 3: how did you make sure that $\eta >0$ for the Quadratically Greedy Linesearch?

Minor:

* line 154: it should be "a direction", as there can be multiple eigenvectors for $\lambda_t$.
* line 139: it would be easier to grasp if there is a short summary of the results of Vaswani et al.

**Limitations:**

See weaknesses and questions. No code is provided.

---

> ### Author Rebuttal · Authors · 2024-08-06
>
> We thank the reviewer for reading this manuscript and providing valuable feedback.
>
> We answer their comments below.
>
> > _"The proposed CDAT tuner has several limitations"_
> - Our goal with this study is to underscore the interplay between sharpness dynamics and stepsize tuners. The CDAT rule serves as a probe to question both the design of learning rate tuners and the study of sharpness dynamics beyond constant learning rates settings. CDAT does stabilize the sharpness dynamics that cause classical tuners to fail (Fig. 1) and seems to induce an automatic warmup (Fig. 5). It also demonstrates the limitations of the current theory on the sharpening and edge of stability phenomena. We agree that CDAT is not yet a mature optimizer but we believe the concepts it introduces will help move the field forward.
>
> > _"How much does CDAT increase the runtime for one step?"_
> - As detailed in Appendix C.3, CDAT required twice the wall-clock time of the constant or scheduled learning rates counterparts. The computation of $g_t^\top H_t g_t$ is done by two forward mode autodiff avoiding the memory cost of Hessian vector products. We leave for future work an optimized implementation of the rule by e.g. computing $g_t^\top H_t g_t$ at fixed intervals.
>
> > _"CDAT replaces the necessity to tune the learning rate with tuning sigma."_
> - The usual selection of the learning rate may largely vary with the optimizer, the architecture and the dataset considered. For example a learning rate of 10^{-3} is generally used for Adam while the learning rate of GD/SGD can vary much more. Though the CDAT rule does not always outperform a constant tuned learning rate, it surprisingly performs well across tasks and *optimizers* for $\sigma=2$.
>
> > _"few experiments are conducted in a stochastic regime […] the results are not very convincing"_
> - Linesearches and other stepsize tuners have indeed generally been tested in a stochastic regime (see also Fig. 14). The potential drawbacks of e.g. linesearches have generally been attributed to small batch sizes [11, Fig. 7]. The analysis of stepsize tuners in a full batch regime shows a different picture: linesearches or greedy methods fail because of phenomena like progressive sharpening. One of our contributions is to question general beliefs about the design of learning rate tuners which may have missed important properties by jumping into the performance in a stochastic regime.
> - Rather than adding yet another optimizer to the long list of existing ones, we preferred to provide numerous experiments to diagnose the benefits and/or challenges (such as adapting the scaling to the batch size, Fig. 7) that arise when adapting the learning rate to the sharpness dynamics.
>
> > _"[...] thus the main motivation for the derivation of CDAT, namely to target EoS, seems not fully convincing in the first place."_
> - We apologize for the confusion. For fixed step size, the typical dynamics of EOS is as follows: at early to intermediate times, training reaches the edge of stability and stabilizes there. For intermediate to late times, the sharpness drops below the edge of stability and training converges. The early time behavior corresponds to the regime where there is the most feature learning/the Hessian changes the fastest; late time corresponds more to the final convergence (see e.g. [24, 25]).
> - The key issue with classical stepsize tuners is that they stay below the edge of stability in the early-intermediate dynamics, where the geometry of the loss landscape is developing the most quickly. This is what leads to the runaway process by which the sharpness keeps increasing, which can sometimes prevent training from even getting to the intermediate/late time dynamics where the Hessian changes slowly, and training converges (see Fig. 3). Therefore stabilizing the early dynamics is key to training success. It remains an open question as to how long it is good to stay at the EOS. We will include a more detailed discussion of this point.
>
> > _"In the Damian et al. paper (ref. [34]), the dynamics are not the same as what you write in (4)"_
> - We wrote down the equations with $\lambda$ instead of $y$ (after converting appropriately) in order to link to the previous discussion of $\lambda$. We switched back to the variable $y$ when analyzing the more detailed model.
>
> > _"Why do you use the squared loss for the CIFAR dataset, instead of the cross-entropy loss?"_
> - We followed Cohen [24, 25] that first analyzed the sharpening and edge of stability phenomena on squared losses. We nevertheless demonstrate the behavior of algorithms in various other settings and losses (see Fig. 1, 5, 8, 12, 19, 21).
>
> > _"Why is Fig. 4 using model (4), and not model (5), if the goal is to illustrate learning rate tuners, where the LR depends also of time (model (5))?"_
> - The dynamics does indeed reflect model (5); thank you for catching that error, we will correct the reference.
>
> > _"Fig. 3: how did you make sure that $\eta>0$ for the Quadratically Greedy Linesearch?"_
> - In Sec. 1, we considered the quadratically greedy rule only with optimizers that did not incorporate a momentum term. In that case, the numerator $-g^\top u$ of the quadratically rule is always positive since the updates $u$ are either the negative gradients $-g$ (for GD) or an element-wise positive scaling of the negative gradients (for RMSPROP). The denominator $u^\top H u$ could be negative only (i) if the Hessian is not positive definite at the updates, (ii) the updates do not belong to the eigenspace of positive eigenvalues. The eigenvalues of the Hessian are generally overwhelmingly positive, and, in practice, we have never observed a negative learning rate. The CDAT rule with $\sigma=1$, that clips the learning rate to positive values can also be seen as a "safe'' quadratically greedy rule.
>
> Thanks for catching the typos that we will correct in our manuscript. We look forward to a fruitful discussion that can further elucidate any concerns.

---

> > ### Comment · Reviewer_jCtW · 2024-08-08
> > **Thank you for rebuttal**
> >
> > Thank you for the detailled rebuttal, and for adressing all of my questions.
> >
> > * Regarding my concerns on practical use: from your rebuttal it seems that we agree that CDAT currently is not intended to be a fully practical optimizer yet, mainly because of several mentioned limitations like increased time per iteration, problems in the minibatch-setting, etc. While this is fine for me, it is indeed a limitation of the paper, because it will require a lot of further engineering and research effort to make it a practical method.
> > * Regarding sensitivity on $\sigma$: indeed $\sigma=2.0$ seems to perform well, but still does not outperform (tuned) GD on 2 out of 4 tasks in Figure 5. My main concern is that a slightly different value of $\sigma=1.94$ can result in quite different loss curves (e.g. Fig. 5, Mixer), suggesting that the CDAT dynamics are sensititve to the choice of $\sigma$.
> >
> >
> > Other replies:
> >
> > >  thus the main motivation for the derivation of CDAT, namely to target EoS, seems not fully convincing in the first place.
> >
> > Thanks for clarifying, I now understood that the main focus is on the initial phase of training. In that case, I agree that line-search methods show a distinct behaviour than GD. I would kindly ask you to stress this in the final version, in order to avoid confusion.
> >
> > > the dynamics are not the same as what you write in (4)
> >
> > Redoing my calculations, I now obtain the correct model matching the one in (4). I recommend to explain this reparametrization in more detail (e.g. in appendix), as your notation does not match the one from Damian et al..
> >
> >
> > I am considering to raise my score, and will do so in the end of the discussion period, after spending more time to read the other reviews/rebuttals.

---

> > > ### Author Response · Authors · 2024-08-09
> > > **Answering additional comments**
> > >
> > > We sincerely thank the reviewer for reading our answers. We answer below their remaining comments.
> > >
> > > > _"Regarding my concerns on practical use: from your rebuttal it seems that we agree that CDAT currently is not intended to be a fully practical optimizer yet, mainly because of several mentioned limitations like increased time per iteration, problems in the minibatch-setting, etc. While this is fine for me, it is indeed a limitation of the paper, because it will require a lot of further engineering and research effort to make it a practical method."_
> > > - We agree with the reviewer. We wanted to point out that there is a path for CDAT to have practical impact beyond being an end-to-end optimizer. Overall it does seem that CDAT does particularly well in choosing a learning rate schedule early in training. This suggests that combining CDAT with methods which handle late time convergence like (1) is a promising avenue. Additionally, CDAT might be combined with scaling ladders: CDAT (or refinements) can find efficient warmup schedules at small to medium scales, and then we may transfer these schedules to larger scales (similar to the findings in (2)). The current extensive set of preliminary results serve as a necessary basis for such refinements.
> > >
> > > > _"Regarding sensitivity on $\sigma$: indeed $\sigma=2$ seems to perform well, but still does not outperform (tuned) GD on 2 out of 4 tasks in Figure 5. My main concern is that a slightly different value of $\sigma=1.94$ can result in quite different loss curves (e.g. Fig. 5, Mixer), suggesting that the CDAT dynamics are sensitive to the choice of $\sigma$"_
> > > - Thanks for raising this point; we will add more detailed analysis of sensitivity to $\sigma$. We find that using an exponential moving average (EMA) in the “edge” estimation tends to smooth out performance across $\sigma$ (see e.g. the stochastic experiments in Fig 7). We have similar experiments in the full batch case (Fig. 5 Mixer) that will be added.
> > > - The fact that a single value (here $\sigma = 2$) can serve as a global scale across optimizers in the full batch setting suggests that testing just a few settings provides a lot of information on the optimal $\sigma$: just below 2, at 2 itself, and just above 2. This contrasts with usual learning rate tuning of e.g. sgd with momentum whose scale varies heavily between problems.
> > >
> > > > _"I would kindly ask you to stress this in the final version, in order to avoid confusion."_, _"I recommend to explain this reparametrization in more detail"_
> > >
> > > Absolutely, we will make these changes for the final version. Thank you for all your feedback in this reviewing process!
> > >
> > > *References*:
> > > (1) The Road Less Scheduled, https://arxiv.org/abs/2405.15682
> > > (2) Why do Learning Rates Transfer? Reconciling Optimization and Scaling Limits for Deep Learning, https://arxiv.org/abs/2402.17457

---

### Official Review · Reviewer_J393 · 2024-07-04

**Soundness:** 4
**Presentation:** 4
**Contribution:** 3
**Rating:** 7
**Confidence:** 4

**Summary:**

The paper investigates the consequences of the sharpness dynamics on step-size tuners' design. In particular, given a learning rate $\eta$, the sharpness exhibits a progressive sharpening phase towards the Edge of Stability (EoS) threshold of $2/\eta$ , where it stays for a large part of training time. First, the authors analyze the sharpness dynamics for two popular step-size tuners (linesearch and quadratically greedy tuners). In both cases, it is observed a significant decrease in learning rate over time, which coincides with a sharpening phase to ever-increasing thresholds. This leads to a suboptimal schedule of ever-decreasing learning rates that might perform better in a single step, but fails to deliver good performances at longer time scales. To fix this vicious cycle, the paper proposes Curvature Dynamics Aware Tuning (CDAT), which takes into consideration the alignment of the gradient with the Hessian, and it is designed to operate "at the edge" through a scalar multiplier $\sigma$. The intuitions are corroborated with simple theoretical analyses on the interplay between sharpness dynamics and learning rate schedules, which (compared to when the learning rate is fixed) have both a time dependence.

**Strengths:**

1. **Novelty of the idea**. Compared to classical analysis (e.g. linear models) where the sharpness $\lambda_t$ is fixed at initialization, in neural networks the sharpness dynamics exhibit the consistent phenomenon of progressive sharpening towards the value of $2/\eta$ (idea crystallized in Cohen et al. 2021). Given that the sharpness also provides a bound on the maximum step size allowed based on the local landscape, there is an interesting interplay between sharpness dynamics and learning rate, especially when the learning rate is varied either with a fixed schedule (e.g. warmup and decay) or with a step-size tuner. Thus, by analyzing step-size tuners, the paper is a step toward understanding this delicate interplay. To my knowledge, this is the first work toward this direction and it's the main strength of the paper.

2. **Clarity**. The paper is exceptionally well-structured and flows nicely. The structure goes back and forth from empirical evidence to the theoretical model which provides intuitive justification. Overall, the paper is also very well-written and self-contained, and it provides all the necessary (and sufficient) context. The experiments are also portrayed schematically and intuitively.

3. **CDAT tuner**. The idea of the new proposed scheduler is simple and effective, and captures many interesting properties of the interplay between sharpness and learning rate, outperforming the base tuners in the full-batch case. Also, the authors put in a little extra engineering to ensure the stability of the optimizers, which makes intuitive sense.

**Weaknesses:**

I am overall strongly in favor of acceptance. However, my score is not higher for the following reasons:

1. **Practical Limitations of CDAT**: I understand that the main purpose of the paper is to diagnose optimization properties at the edge of stability, and to study the interplay between learning rate and sharpness. However, there are a few practical limitations of the proposed tuner CDAT. First, it does not outperform the baseline of a constant learning rate in the practical deep learning use-case of mini-batch optimization, even after tuning the $\sigma$ parameter which is supposed to take into account stochastic batch noise. Furthermore, it has an additional hyperparameter $\sigma$ that has to be tuned. Thus, it loses the advantages of a learning rate tuner in the first place.

2. The authors attribute the lower performances of CDAT for the stochastic regime to (1) the optimal scaling factor is mini-batch dependent, and (2) that the sharpening effect is mitigated in the stochastic regime. However, (1) is not tested. Also, the fact that even after tuning the scaling factor $\sigma$, CDAT underperforms is a partial indication that something else (beyond the fact the stochastic batches lower the sharpness threshold below $2/\eta$) is responsible for the drop in performance. Again, it could be because you need a different $\sigma$ per batch, but this is not experimentally validated. Also, this would further limit the applicability of the tuner.

3. I would appreciate it if the theoretical model from Damial et al (2022)(together with its underlying assumptions) is summarized, at least in the Appendix.

**Questions:**

1. The authors provide an experiment where the tuner uses the exact sharpness instead of CDAT, which takes into account how the update is aligned with the largest eigenvector. What conclusions can be drawn on the learning rate and Hessian interplay? For instance, that at the beginning of training, the updates are not aligned with the leading eigenvector, which allows you to take larger steps (i.e. by increasing $\sigma$)?

2. Varying width and depth. The authors have an experiment (Fig. 18) where either the width or the depth is increased. There is a line of work that studies how hyperparameters (such as the learning rate) transfer from small to large scale (Yang et al, 2022 https://arxiv.org/abs/2203.03466). What are the implications of this paper's results in this context? It is my understanding that [2] can be related/discussed.

These papers seem relevant for the discussion on EoS and its relation to the learning rate, and varying hyperparameters:

[1] Universal Sharpness Dynamics in Neural Network Training: Fixed Point Analysis, Edge of Stability, and Route to Chaos (https://arxiv.org/abs/2311.02076)

[2] Why do Learning Rates Transfer? Reconciling Optimization and Scaling Limits for Deep Learning (https://arxiv.org/abs/2402.17457)

[3] Understanding Gradient Descent on the Edge of Stability in Deep Learning (https://arxiv.org/abs/2205.09745)

**Limitations:**

The authors provide extensive experimental details that make the results fully reproducible. Also, the Appendix provides a lot of interesting ablations, such as varying width and depth, additional base learning rate tuners, and comparisons with commonly used prefixed schedules. In general, I find the suite of experiments very extensive. Finally, I also appreciate that the authors provide an extensive limitation section, setting the boundaries to which aspects of the interplay between sharpness and learning rate dynamics can be captured by CDAT.

---

> ### Author Rebuttal · Authors · 2024-08-06
>
> We sincerely thank the reviewer for the positive feedback and to appreciate the exploratory nature of this work.
>
> Below we answer their comments.
>
> > _"it does not outperform the baseline of a constant learning rate in the practical deep learning use-case of mini-batch optimization [...] it has an additional hyperparameter $\sigma$"_
> - Yes, the mini-batch case suggests that a rule placing the optimizer "on the edge'' needs to take into account stochasticity. We have been first and foremost interested in the full-batch case because the performance of e.g. linesearches reported in the stochastic setting left out the intriguing poor performance in large or full batch cases (see also [31]). While the CDAT rule still requires additional mechanisms to handle stochasticity (the recent schedule-free optimizer (1) may be combined with the proposed rule for example), we preferred providing numerous ablation studies to understand the interplay between sharpness dynamics and learning rate tuners in diverse scenarios.
> - Note that at larger batch sizes, a scaling factor of 2 still works well (Fig. 7 and 20). On the other hand, the behavior of usual tuners such as linesearches suffer from unexpected changes in behavior as the batch size changes: at both small and large batch sizes they can perform worse than constant learning rate counterparts (see e.g. [31]). We hope that the insights provided in the full-batch setting gathered in the provided experiments will help refine the design of learning rate tuners in the stochastic case.
> We agree that in the stochastic case the parameter $\sigma$ still requires to be tuned. On the other hand, we note that in the full batch setting the CDAT rule with a scaling of $\sigma=2$ performs well across different optimizers. This is particularly surprising given that an optimizer such as Adam generally requires a learning rate of $10^{-3}$ while the learning rates of e.g. SGD with momentum vary greatly with the problem.
>
> > _"[the fact that] the optimal scaling factor is mini-batch dependent [...] is not tested"_
> - We would like to point the reviewer to  Fig. 7, which shows that the optimal scaling factor varies with the mini-batch size. Kindly let us know if that doesn't answer the question.
>
> > _"I would appreciate it if the theoretical model from Damian et al (2022)(together with its underlying assumptions) is summarized"_
> - Thank you for the suggestion. We will add a complete introduction to the model of Damian et al (2022) for the final version. The gist of the model is summarized in page 6 line 161.
>
> > _"What conclusions can be drawn on the learning rate and Hessian interplay? [...] the updates are not aligned with the leading eigenvector [...] ?"_
> - One interesting feature of CDAT is that it allows for learning rates to be instantaneously larger than the edge of stability. If the updates are not aligned with the large eigenmodes (as is the case during early iterations), the tuner chooses large step sizes - since the instability won’t matter over a few steps. More generally, the dynamical nature of the alignment allows CDAT to be more flexible and respond to changing curvature in a way that overall leads to stabilization at the EOS without sacrificing optimization.
> - We would also like to highlight Fig. 5 top left panel: placing the optimizer well above the edge ($\sigma=2.5$) does not necessarily lead to divergence in early optimization stages. The optimizers may in fact benefit from a mechanism that ensures no growth in sharpness at the start, beyond thinking whether the optimizer is then unstable according to some quadratic approximation. It is still unclear whether choosing a learning rate rule can actually extract some third order derivative information that drives the sharpness into lower values.
>
> > _"What are the implications of this paper's results in [the] context [of [2]]?"_
> - Thank you very much for the interesting reference! Our goal is somewhat orthogonal to this work, as we aim to harness the sharpening effects to derive an "automatic warmup'', as well as an automatic selection of the learning rate once a relevant warmup phase has cooled down. Yet, the provided reference suggests that a fixed warmup may also transfer across scales. The question is then what shape the warmup must take and what is it controlling. The CDAT rule may give a post-hoc explanation to this question. Our hope is to let a learning rate tuner such as CDAT capture the right warmup shape and peak learning rate in a stochastic regime with small to medium models and then use scaling laws to transfer the found warmup schedule to larger models.
>
> *References*:
> (1) The Road Less Scheduled, https://arxiv.org/abs/2405.15682

---

> > ### Comment · Reviewer_J393 · 2024-08-11
> > **Response**
> >
> > I thank the authors for addressing my concerns and questions.
> >
> > **On the performance of CDAT**: I would like to clarify that I do not find the lower performance of CDAT in the stochastic setting as an inherent weakness of the paper, but in a sense, it is an objective limitation that prevented me from giving an even higher score. However, I appreciate the authors' efforts to run these experiments beyond the setting where the theory applies. It would just have been even greater if CDAT (or a variation of it) was working in more practical settings.
> >
> > > [the fact that] the optimal scaling factor is mini-batch dependent [...] is not tested
> >
> > By mini-batch dependent, I referred to the dependence of every *single* mini-batch, and not the *size* of the batch, as the authors point out in lines 237-239. It would be interesting to see how the sharpness varies on a mini-batch level, i.e. ultimately how much the stochastic noise in the batch affects the curvature across training.
> >
> > I remain in solid favor of acceptance.

---

### Official Review · Reviewer_cxBm · 2024-07-05

**Soundness:** 3
**Presentation:** 3
**Contribution:** 3
**Rating:** 6
**Confidence:** 4

**Summary:**

This papers studies the behavior of several automatic learning rate schedulers in deep learning.  First, the paper studies two classical learning rate tuners - line search, and quadratically greedy (which chooses the learning rate that minimizes a quadratic Taylor approximation in the negative gradient direction).  The paper shows that while these two methods work well on a linear model, they severely outperform fixed-step-size GD on deep learning problems, in the full-batch setting.  The paper explains these results by noting that fixed-step-size GD automatically controls the sharpness along its trajectory, whereas these two classical learning rate tuners do not do this; hence, as training goes on, the sharpness increases and the estimated step sizes decrease.  The paper shows that this intuition can be reproduced in a simplified model of sharpness dynamics that was proposed in a prior work.   The paper then suggests a new automatic learning rate rule called "CDAT" which uses some fixed scaling $\sigma$ of the quadratically greedy schedule; $\sigma=2$ corresponds to a rule which always chooses as a step size that yields no predicted change in loss, under a local quadratic model.  The paper shows that in the full-batch regime, CDAT usually does as well as, or better than, GD with a fixed step size, though not as well as GD with an optimally tuned learning rate schedule.   The paper finds that in the _minibatch_ regime, the classical line search methods work well and CDAT performs worse; the paper explains this by noting that progressive sharpening is attenuated in the minibatch regime.  Finally, the paper shows that the benefits of CDAT can be partially captured by a simplified model of sharpness dynamics.

**Strengths:**

I think the paper will be valuable in shedding light on the use (or lack thereof) of classical learning rate schedulers in deep learning.  The paper is high quality, original, and clear.

**Weaknesses:**

The main weakness of the paper is that it studies optimizers which are not state-of-the-art in the first place, and then does not fully or satisfactorily explain the behavior of these optimizers.  However, these weaknesses are arguably to be expected given that we are still in the very early stages of the theoretical study of optimization in deep learning.  In particular, no other papers contain stronger analyses than the analyses here.

**Questions:**

Figure 16 seems to have the same figure accidentally repeated twice.

I appreciate the ablation on MNIST, but I am not sure if the findings will transfer to other settings.  For example, I am skeptical about the finding that the efficacy of CDAT goes away with large weight decay.

---

> ### Author Rebuttal · Authors · 2024-08-06
>
> We thank the reviewer for reading our paper and providing valuable feedback.
>
> We answer below their comments.
>
> > _"it studies optimizers which are not state-of-the-art in the first place"_
>
> - Could the reviewer clarify what optimizer do they have in mind? We already added additional learning rate tuners (Polyak stepsizes and hypergradient descent) in Fig. 13 and plan to add also recent tuners such as Prodigy (1) or DoG (2). In practice Adam and its variants are generally still considered as the main optimizers used in practice, hence we provided experiments with those.
>
> > _"Figure 16 seems to have the same figure accidentally repeated twice."_
>
> - Thanks for catching the mistake on Fig. 16, we corrected this. The rightmost figure offered a similar conclusion as detailed in the caption.
>
> > _"I am skeptical about the finding that the efficacy of CDAT goes away with large weight decay."_
>
> - As pointed out by J393, a future step will be to understand the interplay between stepsize tuners and sharpness dynamics in terms of scaling laws (using the findings of (3)). In particular, we agree that ablation studies as done in Fig. 18 will benefit from placing ourselves in a rich feature learning limit.
>
> _References_:
> (1) Prodigy: An expeditiously adaptive parameter-free learner, https://arxiv.org/abs/2306.06101
> (2) Dog is sgd’s best friend: A parameter-free dynamic step size schedule, https://arxiv.org/abs/2405.15682
> (3) Why do Learning Rates Transfer? Reconciling Optimization and Scaling Limits for Deep Learning, https://arxiv.org/abs/2402.17457

---

> > ### Comment · Reviewer_cxBm · 2024-08-09
> > **response**
> >
> > > Could the reviewer clarify what optimizer do they have in mind?
> >
> > Sorry, I was vague -- my point was that CDAT is not a state-of-the-art optimizer, which limits the significance studying it (especially in a way that leaves a lot of questions open).

---

### Official Review · Reviewer_7YAk · 2024-07-08

**Soundness:** 3
**Presentation:** 3
**Contribution:** 3
**Rating:** 6
**Confidence:** 4

**Summary:**

The paper proposes a novel learning rate tuning method, CDAT, that leverages the largest Hessian eigenvalue information during training. To illustrate the feedback loop between learning rate selection and sharpness dynamics, and to emphasize the importance of stepping on the edge of stability, the authors introduce a toy model. In the full batch setting, CDAT is shown to outperform constant learning rate schedules. Furthermore, under the mini-batch regime, the authors demonstrate that CDAT can naturally discover a warmup schedule in specific scenarios.

**Strengths:**

1. The paper is well-organized and easy to understand.
2. The authors offered very detailed hyperparameter choices used in the paper.

**Weaknesses:**

1. Theoretical ground is not solid
- For stochastic gradient descent, it was known that the correct measure of sharpness was the trace of Hessian instead of the largest eigenvalue [1], so using the largest eigenvalue of Hessian for SGD cases is not reasonable.
- For adaptive optimizers like Adam, the correct quantity to measure is the pre-conditioned Hessian [2], the largest eigenvalue of which is usually very large at initialization. In Fig 8 Adam on the edge, the initial learning rate is very large, which is most likely because the authors used Hessian instead of pre-conditioned Hessian.
- For cross-entropy loss, [3] has shown that the sharpness will reach $2/\eta$ and then come down later. This is not covered anywhere in the paper.
- At early stage of training, using $\eta \sim 2 / \lambda_{max}$ does not mean it will remains on the EoS. As the [4, 5] have shown, there could be a catapult phase for $\eta > 2 / \lambda_{max}$, or depending on the initialization, the sharpness might decrease or increase at an early stage.

[1] Lei Wu, Weijie J Su, https://proceedings.mlr.press/v202/wu23r.html

[2] Jeremy M. Cohen, et al. https://arxiv.org/abs/2207.14484

[3] Jeremy M. Cohen, et al. https://arxiv.org/abs/2103.00065

[4] Aitor Lewkowycz, et al. https://arxiv.org/abs/2003.02218

[5] Dayal Singh Karla, et al. https://arxiv.org/abs/2311.02076

2. Experimental Results are not good:
- It takes 500 epochs to train a ResNet-50 to $<80\%$ on CIFAR-10
- For most real-world settings, CDAT can not outperform existing simple cosine-annealing schedules.

**Questions:**

Could the authors address my concerns in the weakness section?
Please correct me if I misunderstood anything from the paper. I am willing to discuss this further.

**Limitations:**

Yes

---

> ### Author Rebuttal · Authors · 2024-08-07
>
> We thank the reviewer for their detailed feedback, and address their comments here.
>
> > _"Theoretical ground is not solid"_
> - The experiments and models focus on the full batch regime just as Cohen [24, 25] did to uncover the edge of stability phenomenon. The theory is grounded in the full batch regime following previous work [34]. We clearly point out the limitations of the model in a stochastic regime referring to [24, 32, 33] (we thank the reviewer for the additional reference that we will add).
>
> > _For [SGD], [...] the correct measure of sharpness [is] the trace of Hessian instead of the largest eigenvalue [1], so using the largest eigenvalue [...] for SGD [...] is not reasonable.''_
> - We would like to start by clarifying what we believe is a critical misunderstanding. The CDAT rule **does not use the largest eigenvalue of the Hessian**. In fact as demonstrated by [24, Appendix F], and further explored in Fig. 15, using the exact sharpness does not provide gains unless the scaling is much larger than 2 (3 in that case). The CDAT rule **incorporates the alignment of the gradient with the Hessian** as reviewer J393 also identified. That's what the model in Sec. 3.2 points out too.
> - The trace of the Hessian indeed controls the EoS in highly stochastic settings;  the largest eigenvalue controls stability due to first moments, while the trace controls second moments (3). The trace becomes more important at later times, e.g. during learning rate decay. This means that in SGD settings, there are ranges of batch sizes/stages of training where the largest eigenvalue can still control stability/EOS behavior ([24, Fig. 24];  (4, Fig. 5)). The CDAT tuner seems to have the most beneficial effect in the early stage of training (to induce some warm-up). In this first work we designed CDAT around these first moment effects and focused on empirical studies, but we hope to incorporate late time stochastic effects into future work.
>
> > _"For adaptive optimizers [...], the correct quantity [...] is the pre-conditioned Hessian [...]. In Fig 8 [...], the initial learning rate is very large, which is [...] because the authors used Hessian instead of pre-conditioned Hessian.''_
> - The CDAT rule takes care of this implicit preconditioning. Both the numerator and denominator depend on the *update* $u$. For pre-conditioned optimizers, $u = -P^{-1} g$, where $g$ is the gradient, and $P$ is the diagonal preconditioner. The reciprocal of the CDAT rule then reads
> $$
> \frac{u^\top H u}{-u^\top g}  = \frac{g^\top P^{-1} H P^{-1} g}{g^\top P^{-1} g}.
> $$
> If we were to maximize the above ratio, we would get
> $$
> \max_g \frac{g^\top P^{-1} H P^{-1} g}{g^\top P^{-1}g} = \max_v \frac{v^\top P^{-1/2} H P^{-1/2} v}{\||v||^2} = \lambda_{\max}(P^{-1/2}H P^{-1/2}),
> $$
> and the CDAT rule "on edge" would then be $2/\lambda_{\max}(P^{-1/2}H P^{-1/2})$, the edge of stability of adaptive gradient methods without momentum [25] (again the CDAT rule does not $\lambda_{\max}$ but incorporates the alignment).
> - The initial large learning rate in Fig. 8 is likely due to the fact that the preconditioner itself is highly variable in the very first iterations.
>
> > _"For cross-entropy loss, [...] the sharpness will reach $2/\eta$ and then come down later […] using $\eta\sim 2/\lambda$ does not mean it will remain on the EoS [...] depending on the initialization, the sharpness might decrease or increase at an early stage.''_
>
> - The curvature dynamics for **constant learning rates** is indeed complex. However, the complexities pointed out by the reviewer (that will be added) need to be revisited in the context of **variable learning rates defined through a learning rate tuner**. Learning rate tuners introduce *closed loop feedback effects*, and so even more complex behavior. To revisit these results, a first set of experimental results need to be laid down and that's what we present. The complexities pointed out by the reviewer do not seem crucial to observe the failure of classical learning rate tuners. On the other hand, the stabilization mechanisms at EoS (captured by the models of [34], and observed in many nonlinear models trained at large batch size during some appreciable fraction of training) can provide first insights on the closed loop feedback effects. That's what the CDAT rule puts to the test. CDAT provides a closed loop feedback that *encourages* and is *compatible with* EoS dynamics - but does not put a hard-constraint towards EoS. Our experiments with CDAT show some novel behaviors. For example, in Fig. 6 $\lambda_{\max} \eta$ stabilizes below 2 for GD, while for CDAT $\lambda_{\max} \eta$ remains slightly above 2. We also observe that using the CDAT rule, the sharpness may even decrease at later times (Fig. 6). These empirical findings are new and crucial to put the EoS studies into practical use.
>
> > _"Experimental results are not good.''_
> - See main rebuttal. We agree that CDAT is not yet a mature solver. We preferred to focus on an extensive set of experiments analyzing the idea rather than adding yet a new optimizer whose performance would be left to be diagnosed by peers. None of the references provided by the reviewer provided a new state of the art solver, neither did they propose new methodologies to design new solvers.
>
> Please let us know if we can clarify our thinking further; we look forward to a fruitful dialogue with you.
>
> *References*:
> - (1) The Road Less Scheduled, https://arxiv.org/abs/2405.15682
> - (2) Second-order regression models exhibit progressive sharpening to the edge of stability, https://arxiv.org/abs/2210.04860
> - (3) High dimensional analysis reveals conservative sharpening and a stochastic edge of stability, https://arxiv.org/abs/2404.19261
> - (4) SAM operates far from home: eigenvalue regularization as a dynamical phenomenon, https://arxiv.org/abs/2302.08692

---

> > ### Comment · Reviewer_7YAk · 2024-08-08
> >
> > I thank the authors for providing detailed explanations. After reading through the rebuttal, my concerns are mostly solved, and I
> > have raised my score. Still I have some questions/concerns I hope the authors can address:
> >
> > 1. I still think that the performance of CDAT is a weakness. Also, the extra computing cost is not that small. I wonder if an estimate for every $t$ (like 1000) steps would help[1], but this might lead to instability issues as the model is running "on the edge".
> >
> > 2. For Figure 5, have the authors tried comparing CDAT with, say, a warmup and then a cosine schedule? GD might be fine, but it is believed Adam needs some warmup [2] to perform.
> >
> > 3. Still related to Adam, [3] showed that at an early stage, the pre-conditioned hessian has extremely large eigenvalues, which seems to be in contrast with the Adam learning rate found by CDAT in Figure 8. I know that the authors' work is not focusing on top-eigenvalues but this is still worry me a bit.
> >
> > [1] Sophia: A Scalable Stochastic Second-order Optimizer for Language Model Pre-training, H. Liu, et al., https://arxiv.org/abs/2305.14342
> >
> > [2] On the Variance of the Adaptive Learning Rate and Beyond, L. Liu, et al., https://arxiv.org/abs/1908.03265
> >
> > [3] Adaptive Gradient Methods at the Edge of Stability, Jeremy M. Cohen, et al. https://arxiv.org/abs/2207.14484

---

> > > ### Author Response · Authors · 2024-08-09
> > > **Answering additional comments**
> > >
> > > We sincerely thank the reviewer for reading our answers. We answer their remaining comments below.
> > >
> > > > _"I still think that the performance of CDAT is a weakness. Also, the extra computing cost is not that small. I wonder if an estimate for every $t$  (like 1000) steps would help[1], but this might lead to instability issues as the model is running "on the edge"."_
> > > - We kindly ask the reviewer to see the answer we also provided to reviewer jCtW on this point: there is a path for CDAT to have practical impact beyond being an end-to-end optimizer. The current extensive set of preliminary results serve as a necessary basis for such refinements.
> > > - That said, the importance of "instantaneous changes in the edge" is indeed very interesting, and our work suggests an approach which measures curvature less frequently may be viable. Our experiments integrating the exponential moving average (EMA) in the evaluation of the rule provides some evidence about smoothing curvature information over time. In the stochastic setting, we used this EMA by necessity, and Figure 5 shows that using EMA = 0.9 prevents divergence for $\sigma>2$. We conducted additional experiments on the use of EMA in the full batch setting and will include them in our revision. These results encourage us that there are stable and useful variations which subsample curvature in time.
> > >
> > > > _"For Figure 5, have the authors tried comparing CDAT with, say, a warmup and then a cosine schedule? GD might be fine, but it is believed Adam needs some warmup [2] to perform."_
> > > - Comparison with schedules are presented in Fig. 19, 20, 21 (detailed experimental setup is presented in Appendix C.5). These plots raise additional questions: should we let the sharpness drive the dynamics as in CDAT, or should we let the warmup drive the sharpness. That's the reason we pointed out some holes in the current EoS literature to harness the sharpness dynamics in the favor of the optimizer.
> > >
> > > > _"Still related to Adam, [3] showed that at an early stage, the pre-conditioned hessian has extremely large eigenvalues, which seems to be in contrast with the Adam learning rate found by CDAT in Figure 8. I know that the authors' work is not focusing on top-eigenvalues but this still worries me a bit."_
> > > - Thanks for re-iterating this point; we dug further into the experiments. It turns out that at initialization (step 0) we logged the learning rate to 1. This created an artifact in the plot. We provide below the detailed learning rates for Adam for Fig. 5 and Fig. 8. We sincerely thank the reviewer for insisting on an explanation, and will correct the paper on this point.
> > > - We note that the initial steps still tend to have larger learning rates. This is because we do not have the correspondence $\eta = 2/\lambda_{\max}$ at early steps, because CDAT is not necessarily aligned with the largest eigenvalue. This is particularly true for the very first step since there is no reason for the gradient to be aligned with the preconditioned hessian at that step. Large learning rate at early times can occur if the gradient has more weight in smaller eigendirections. After a few iterations, the largest eigenvalue of the preconditioned Hessian appears to come down to moderate values (see also e.g. Fig. 12).
> > > - What we found remarkable too is how learning rate tuners may completely change the sharpness dynamics found previously. Adam on Fig. 5 is a good example of this.
> > >
> > > ---------------------
> > > ---------------------
> > >
> > > **Adam on edge ($\sigma$=2) full batch details** (Fig. 5)
> > >
> > > | Step | Learning Rate | Precond. Hessian sharpness |
> > > | ---- | ------------- | -------------------------- |
> > > | 1 | 2.80e-02 | 7.87e+06 |
> > > | 2 | 1.77e-04 | 2.50e+05 |
> > > | 3 | 2.04e-04 | 2.00e+05 |
> > > | 4 | 2.06e-04 | 2.27e+05 |
> > > | 5 | 2.14e-04 | 2.96e+05 |
> > > | 6 | 2.23e-04 | 2.25e+05 |
> > > | 7 | 2.29e-04 | 1.05e+05 |
> > > | 8 | 2.31e-04 | 6.62e+04 |
> > > | 9 | 2.74e-04 | 5.92e+04 |
> > > | 10 | 2.77e-04 | 5.72e+04 |
> > > | 11 | 2.84e-04 | 6.61e+04 |
> > > | 12 | 2.88e-04 | 8.71e+04 |
> > > | 13 | 2.99e-04 | 9.82e+04 |
> > > | 14 | 2.98e-04 | 1.03e+05 |
> > > | 15 | 3.08e-04 | 9.70e+04 |
> > > | 16 | 3.08e-04 | 8.80e+04 |
> > > | 17 | 3.14e-04 | 8.18e+04 |
> > > | 18 | 3.12e-04 | 8.72e+04 |
> > > | 19 | 3.19e-04 | 8.89e+04 |
> > >
> > > --------------------------------
> > >
> > > **Best adam on edge stochastic case batch size 256 (Fig. 8)**
> > > (we don't have the largest eigenvalue of the full batch preconditioned hessian in this case as it is too expensive to compute).
> > >
> > > | Epoch | Learning Rate |
> > > | -------  | ---------------- |
> > > | 4.0 | 5.18e-04 |
> > > | 8.0 | 1.22e-03 |
> > > | 12.0 | 1.25e-03 |
> > > | 16.0 | 9.84e-04 |
> > > | 20.0 | 9.46e-04 |

---

> > > > ### Comment · Reviewer_7YAk · 2024-08-09
> > > >
> > > > Thank the authors for the very detailed explanation. Now I think this work should be accepted, and I have updated the score accordingly.
> > > >
> > > > The learning rate pattern for Adam surprised me a bit. The authors offered a very different view from the current understanding of sharpness dynamics and how to choose a learning rate based on sharpness values. I personally believe this is a very interesting direction worth digging deeper into.

---

### Author Rebuttal · Authors · 2024-08-06

We sincerely thank the reviewers for reading our paper carefully, and providing numerous insightful comments.

We answer each reviewer's comments separately and provide here some answers to the main common comment.

> "Experimental results are not good." (Reviewer 7YAk)
> "it does not outperform the baseline" (Reviewer J393)
> "the results are not very convincing" (Reviewer jCtW)

- The purpose of this paper is to underscore the interplay between sharpness dynamics and learning rate tuners through an extensive set of experiments and a simple rule (CDAT) playing the role of a probe. Sharpness dynamics are best understood in the full batch setting following the earlier work of Cohen [24, 25]. Such a setting reveals unexpected behaviors of classical learning rate tuners (Fig. 1, 3), questioning preconceived beliefs [9, 11]. The proposed CDAT rule is used as a **diagnostic tool** (as reviewer J393 points out) to examine whether the sharpening/edge of stability dynamics (usually studied for constant learning rate) can help designing more efficient learning rate tuners. The experiments in the full batch setting reveal that, across *optimizers* and tasks, a simple scaling of $\sigma=2$ ("on edge") provides remarkable performance, at least much better than usual learning rate tuners.
- Experiments in the stochastic regime reveal, in full transparency, additional challenges. We preferred to present extensive experiments (21 figures) showing both opportunities and pitfalls of a simple idea (placing the optimizer "on edge'') rather than focusing on pure performance. The pure performance viewpoint in a stochastic regime may have led to some misunderstandings about the efficiency of e.g. linesearch methods in deep learning [9, 11] that did not carefully diagnose the separate effects of loss landscape and stochasticity.
- Recent learning rate tuners (Prodigy (1), schedule-free (2), DoG (3), etc…) have also focused solely on the issue of stochasticity motivated by the classical online learning framework. For all these recent optimizers, a warmup is necessary, which the theory cannot explain.  Our work takes an orthogonal approach: focus on a non-stochastic regime from an empirical viewpoint to understand the role of loss landscape. We may then understand how warm-up can be induced by closed-loop feedback effects on sharpness and learning rate dynamics.
- Finally, we are unaware of other work putting the recent findings in sharpness dynamics of optimizers into practice (though the reference (4) provided by reviewer J393 can have practical impacts on scaling laws). The CDAT rule sheds light on the practical benefits of the study of edge of stability/sharpening that drew much attention in past years. It also highlights the importance of studying such dynamics in closed loop feedback scenarios such as the ones induced by learning rate tuners.

_References_:
(1) Prodigy: An Expeditiously Adaptive Parameter-Free Learner, https://arxiv.org/abs/2306.06101
(2) The Road Less Scheduled, https://arxiv.org/abs/2405.15682
(3) DoG is SGD's Best Friend: A Parameter-Free Dynamic Step Size Schedule, https://arxiv.org/abs/2302.12022
(4) Why do Learning Rates Transfer? Reconciling Optimization and Scaling Limits for Deep Learning, https://arxiv.org/abs/2402.17457

---

### Comment · Area_Chair_ukxU · 2024-08-08

Just a friendly reminder to the reviewers to acknowledge the author's rebuttal, so that the discussion period can be used efficiently.

---

### Decision · Program_Chairs · 2024-09-25

**Decision:**

Accept (poster)

**Comment:**

The paper examinates a simple idea: choosing an adaptive learning  to be always `on the edge' instead of in the stable regime as is usually done. The paper makes some interesting observations, and proposes a clean and simple explanation for them. All reviews are in agreement for acceptance after the discussion period.